# Ageing is associated with disrupted reinforcement learning whilst learning to help others is preserved

Jo Cutler [1,2,3✉], Marco K. Wittmann [2,3], Ayat Abdurahman[2,3,4], Luca D. Hargitai [3], Daniel Drew[2,3,5], Masud Husain [2,3,5] & Patricia L. Lockwood [1,2,3,6✉]

Reinforcement learning is a fundamental mechanism displayed by many species. However, adaptive behaviour depends not only on learning about actions and outcomes that affect ourselves, but also those that affect others. Using computational reinforcement learning models, we tested whether young (age 18–36) and older (age 60–80, total n = 152) adults learn to gain rewards for themselves, another person (prosocial), or neither individual (control). Detailed model comparison showed that a model with separate learning rates for each recipient best explained behaviour. Young adults learned faster when their actions benefitted themselves, compared to others. Compared to young adults, older adults showed reduced self-relevant learning rates but preserved prosocial learning. Moreover, levels of subclinical self-reported psychopathic traits (including lack of concern for others) were lower in older adults and the core affective-interpersonal component of this measure negatively correlated with prosocial learning. These findings suggest learning to benefit others is preserved across the lifespan with implications for reinforcement learning and theories of healthy ageing.

[1] Centre for Human Brain Health and Institute for Mental Health, School of Psychology, University of Birmingham, Birmingham, UK. [2] Wellcome Centre for Integrative Neuroimaging, University of Oxford, Oxford, UK. [3] Department of Experimental Psychology, University of Oxford, Oxford, UK. [4] Department of Psychology, University of Cambridge, Cambridge, UK. [5] Nuffield Department of Clinical Neurosciences, University of Oxford, Oxford, UK. [6] Christ Church, University of Oxford, Oxford, UK. ✉email: J.L.Cutler@bham.ac.uk; P.L.Lockwood@bham.ac.uk

Learning associations between actions and their outcomes is fundamental for adaptive behaviour. To date, the majority of studies examining reinforcement learning have tested how we learn associations between actions and outcomes that affect ourselves, and largely focused on these processes at a young age, both in humans and other species[1–5]. However, such self-relevant learning may be computationally separable from learning about actions that help other people. Studies suggest slower learning of associations between actions and outcomes when they are about[6] or affect others[7], henceforth referred to as 'prosocial learning'.

Senescence is associated with a multitude of changes including declines in cognitive functioning and perception, but perhaps the preservation of affective processing and social cognitive abilities[8–10]. However, less is known about how ageing affects social behaviour, despite the critical importance of this question. Social isolation has been found to be as damaging to physical health as smoking or excessive drinking[11]. Social behaviours that benefit others—prosocial behaviours—are vital for maintaining social bonds and relationships[12] across the lifespan. In addition to the benefits for others, prosociality has been linked to improved life satisfaction[13], mental wellbeing[14] and physical health[15] for the person being prosocial, all of which could contribute to healthy ageing. A key aspect of prosocial behaviour is the ability to learn associations between our own actions and outcomes for other people[7]. Here, we use computational models of reinforcement learning in young and older participants to examine the mechanisms that underpin self-relevant and prosocial learning and associations with healthy individual differences in socio-cognitive ability.

Reinforcement Learning Theory (RLT) provides a powerful framework for understanding and precisely modelling learning[16]. In RLT, prediction errors signal the unexpectedness of outcomes and affect the choices we make in the future. The influence that prediction errors have on choices can be modelled individually through the learning rate, which quantifies the effect of past outcomes on subsequent behaviour. The plausibility of reinforcement learning as a core biological mechanism for action–outcome associations is bolstered by our understanding of neurobiology, with prediction errors encoded by single neurons in the ventral tegmental area[17]. Although essential for successful adaptive behaviour, several studies suggest that our propensity for reinforcement learning declines in later life[10]. Compared to younger adults, older adults show learning impairments particularly when action–outcome associations are probabilistic[18] or reverse[19]. Age-related declines in learning ability have been linked to functional and structural changes in frontostriatal circuits[20,21] and dopamine transmission[18,22], which shows a significant age-related decrease[23–25] and has a key role in coding prediction errors[2,26,27]. Indeed, one study showed that administering L-DOPA, a dopamine precursor, to older adults increased their learning rate[28]. Therefore, if reinforcement learning in general declines in older age, we would hypothesise lower learning rates for both self-relevant and prosocial learning in older, compared to younger, adults.

Alternatively, prosocial learning may depend not only on our learning ability but also our motivation to help others[29]. Results from experiments using economic games to measure prosociality have found that older adults tend to be more generous[30,31]. There is also evidence of an age-related increase in charitable donations to individuals in need[32]. At work, older adults engage in more prosocial behaviours than younger adults, according to both self-report data and colleagues' ratings[33]. Finally, self-reported altruism and decisions to donate to others have been shown to increase with age[34]. However, one limitation of these studies is that the paradigms often place self and other reward preferences in conflict. Money for the other person depends on less money for oneself. Moreover, older adults generally have higher accumulated wealth, which would be an important confound in studies of monetary exchange[35]. Prosocial learning avoids this confound by separating outcomes for oneself from outcomes for others. If older adults do indeed value outcomes for others more than young adults, we might expect that whilst self-relevant learning declines with ageing, prosocial learning could be preserved. Comparing young and older adults on self-relevant and prosocial learning provides an opportunity to dissociate how possible age-related changes in cognitive ability and social behaviour impact on learning.

While studies point to potential group differences between young and older adults, there is also substantial individual variability in concern for others, which can be captured by trait measures. Decreased prosocial behaviour and altered self and other reward processing are key features of psychopathic traits[36,37]. Psychopathic traits are characterised by dysfunctional affective-interpersonal features[38,39] and also include an antisocial behavioural dimension[40]. Levels of these traits range from clinically defined psychopathy, a severe personality condition linked to poor life outcomes, violence and criminality[41–43], to lower subclinical levels of psychopathic traits in the general population. This conceptualisation of psychopathy as dimensional, rather than categorical, reflects the Research Domains of Criteria (RDoC) approach to psychiatry[44]. Evidence of similar behavioural and neural profiles between community samples with high levels of psychopathic traits and those with clinical diagnoses of psychopathy[45] is consistent with this RDoC approach. Self-report measures of psychopathic traits mirror the latent structure of clinical psychopathy measures as comprising socio-emotional and behavioural dimensions but are distinct from a clinical diagnosis of psychopathy. These subclinical self-report measures, such as the Self-Report Psychopathy Scale (SRP)[40] used here, can be administered in behavioural studies to index the range of scores found in the general population. The SRP captures psychopathic traits on two dimensions labelled: 'affective-interpersonal' (lack of empathy and guilt) and 'lifestyle-antisocial' (impulsive and antisocial behaviours)[40].

Intriguingly, preliminary evidence suggests that ageing may also be associated with changes in self-reported psychopathic traits[46], which could have important implications for our understanding of an ageing population. In community samples, ageing is associated with a decrease in the socio-emotional and behavioural dimensions of the SRP[47]. These studies highlight the importance of assessing how differences in self-reported psychopathic traits could map on to differences in prosocial behaviours.

Taken together, previous research supports opposing hypotheses for how ageing is associated with self-relevant and prosocial reinforcement learning. On the one hand, evidence suggests that older adults should be impaired at learning, regardless of the recipient, consistent with ageing-related declines in learning ability and dopamine transmission. On the other hand, potential increases in valuing outcomes for others in older, compared to younger, adults would predict preserved prosocial learning ability but reduced self-relevant learning ability. Finally, we expected variation in self-reported psychopathic traits to be associated with learning for others but not self in both age groups.

To distinguish between these competing hypotheses, we tested 75 young (aged 18–36, mean = 23.07, 44 females) and 77 older (aged 60–80, mean = 69.84, 40 females) adults carefully matched on gender, years of education and IQ test performance. Participants completed a probabilistic reinforcement-learning task (Fig. 1) designed to separate self-relevant (rewards for self) from prosocial learning (rewards for another person), as well as controlling for the general valence of receiving positive outcomes (rewards for neither self nor other).

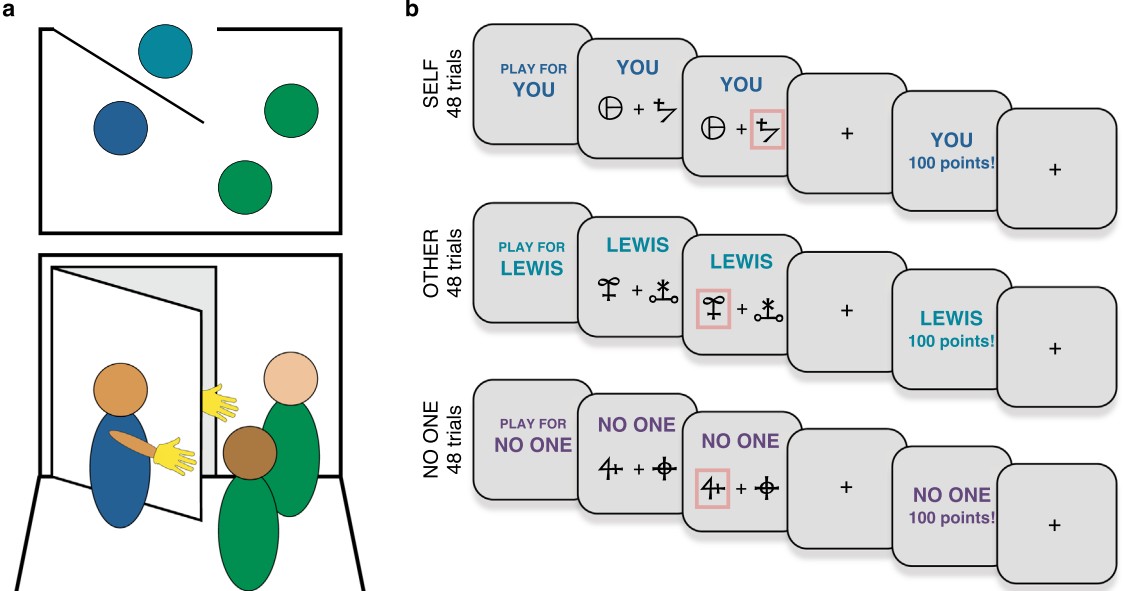

**Fig. 1 Prosocial learning task and social role assignment. a** The role-assignment procedure involved the participant (dark blue), confederate (light blue) and two experimenters (green). Top: from above showing the positioning of the participant and two experimenters inside the testing room, and the confederate on the other side of the door. Bottom: the participant and confederate wore a glove to disguise their identity and waved to each other from either side of the door. Participants were instructed that they would be assigned to roles of 'Player 1' and 'Player 2', but the participant was always assigned to be Player 1. After this procedure, participants were informed that they would play a game where they could gain rewards for themselves, the other participant (Player 2) or neither participant. They were told that Player 2 would not play the same game for them, and that Player 2 would not know that they may receive an additional bonus based on the choices the participant made. This meant that participants' choices were made anonymously and should not be affected by reputational concerns. **b** Participants performed a reinforcement-learning task ('prosocial learning task'), in which they had to learn the probability that abstract symbols were rewarded to gain points. At the beginning of each block, participants were told who they were playing for, either themselves, for the other participant, or in a condition where no one received the outcome. Points from the 'self' condition were converted into additional payment for the participant themselves, points from the 'other' condition were converted into money for Player 2, and points from the 'no one' condition were displayed but not converted into any money for anyone.

The detailed model comparison showed that a computational model with separate learning rates best explained how people learn associations for different recipients (Fig. 2). Young adults were faster to learn when their actions benefitted themselves, compared to when they helped others. Strikingly, however, this was not the case for older adults, who showed a relative increase in the rate at which they learnt about actions that helped others, compared to themselves (Fig. 3a, b). Older adults had significantly reduced levels of self-reported psychopathic traits (mean = 21.09) compared to younger adults (mean = 24.36) and in older adults, lower self-reported psychopathic trait scores correlated with prosocial learning rates (Fig. 4a, b). These effects were not explained by individual differences in IQ test performance, memory or attention abilities. Overall, we show that despite age-related disruption to self-relevant learning, prosocial learning was similar between young and older adults. Ageing was also associated with a decline in self-reported psychopathic traits. These findings suggest learning how our actions help others is preserved across the lifespan.

## Results
We analysed the behaviour of 75 younger adults and 77 older adults who completed the probabilistic reinforcement-learning task (Fig. 1b), neuropsychological tests, and a self-report measure of psychopathic traits (see 'Methods'). To ensure comparability, older adults with dementia, as diagnosed by Addenbrooke's Cognitive Examination (ACE)[48], were not included in the study. The two age groups were matched on gender ($\chi^2(1) = 0.45$, $P = 0.50$) and did not differ in years of education or IQ test performance (Supplementary Table 1). IQ test performance was quantified using age-standardised scores on the Wechsler Test of

Adult Reading (WTAR)[49]. We conducted additional analyses controlling for IQ test performance (standardised WTAR score, measured for young and older adults), and memory and attention (memory and attention subscales of the ACE, older adults only). These control analyses showed that our results are not accounted for by IQ test performance or executive function (see 'Methods' and Supplementary Information).

**Learning occurs for all recipients for both age groups**. We first examined whether participants were able to learn for all three recipients to validate their ability to complete the task. We quantified performance as selecting the option associated with a high chance of receiving the reward. Participants in both age groups were able to learn to obtain rewards for themselves, another person and no one. This was demonstrated through average performance above chance level (50%; all $t$s > 15.46, all $P$s < 0.001) and a significant effect of trial number in predicting trial-by-trial performance (all $Z$s > 4.48, $P$s < 0.001) for each separate recipient and age-group combination.

**Learning rate depends on who receives the reward**. Next, to quantify learning, we used computational models of reinforcement learning to estimate learning rates ($\alpha$) and temperature parameters ($\beta$), key indices for the speed by which people update their estimates of reward, and the precision with which they make choices, respectively. Models were fitted using a hierarchical approach and compared using Bayesian methods[50,51], with model selection based on the exceedance probability and integrated Bayesian Information Criterion (BIC; see 'Methods'). We tested multiple models that varied with respect to whether learning could be

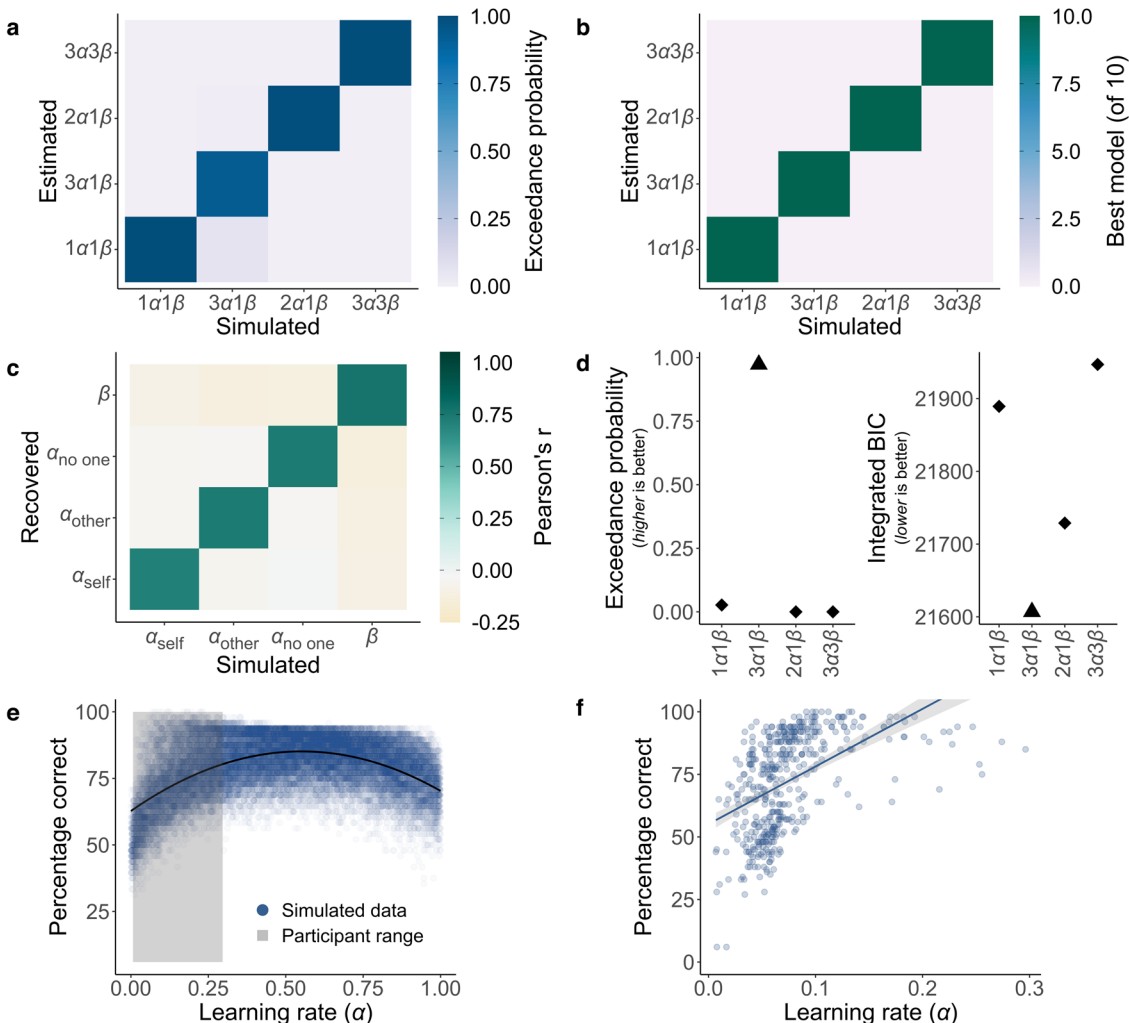

**Fig. 2 Model identifiability, parameter recovery, model comparison and optimal learning rate. a** Model identifiability average exceedance probability confusion matrix and **b** model identifiability best model selection confusion matrix. Data were simulated from 150 synthetic participants with each of our four models then Bayesian model selection was applied, and this procedure was repeated ten times. Identifiability is shown by strong diagonals. **c** Parameter recovery was performed on data simulated by the winning $3\alpha1\beta$ model from 1296 synthetic participants. The confusion matrix represents correlations between simulated and fitted parameters. Stronger colours show higher values and high values on the diagonal show parameters can be recovered. **d** The $3\alpha1\beta$ model (triangle) is the best model on both exceedance probability and integrated Bayesian Information Criterion (BIC) fit measures. **e** Average percentages of correct choices (high probability of reward option) associated with 30,000 simulated $\alpha$ values (10,000 synthetic participants, three recipient conditions; blue dots) show that an optimal learning rate is approximately 0.55 in this task. The range of $\alpha$ values for our participants was below this peak (grey shading), such that a higher learning rate was associated with better performance. **f** Correlation between percentage correct and learning rate across participants. There was a significant Spearman's Rho correlation between learning rate and accuracy ($r_{s(150)} = 0.58$ [0.46, 0.68], $P <$ 0.001) (see Supplementary Table 3 for each separate age group and recipient combination; $r_s$ in all cases > 0.46, $P$s < 0.001), $n = 150$ (75 young, 75 older). The shaded area represents 95% confidence interval. Source data are provided as a Source Data file.

explained by shared or separate free parameters across recipients (self, other no one). Based on our previous results[7], we examined whether shared or separate learning rates, in particular, resulted in a better model fit. We used four candidate models:

(i) $1\alpha1\beta$: one $\alpha$ for all three recipients and one $\beta$ for all three recipients
(ii) $3\alpha1\beta$: $\alpha_{self}$, $\alpha_{other}$ & $\alpha_{no one}$, one $\beta$
(iii) $2\alpha1\beta$: $\alpha_{self}$ & $\alpha_{not\text{-}self\ [other\ +\ no\ one]}$, one $\beta$
(iv) $3\alpha3\beta$: $\alpha_{self}$, $\alpha_{other}$, $\alpha_{no\ one}$, $\beta_{self}$, $\beta_{other}$ & $\beta_{no\ one}$ (see Supplementary Table 2)

Initially, we aimed to establish that both our experimental schedule and our models were constructed in a way that allowed us to disentangle recipient-specific learning rates. To this end, we created synthetic choices using simulations based on each of our four models (see 'Methods'). We fitted the models to the data and

assured that the best-fitting model was the one that had been used to create the data. In such a way, we established model identifiability, both when considering the exceedance probability (Fig. 2a and see 'Methods') and the number of times a model was identified as the best one (Fig. 2b). As a second prerequisite for testing for agent-specific learning rates, we performed parameter recovery using our key model of interest, the $3\alpha1\beta$ model. Over a wide parameter space, we were able to recover the parameters underlying our choice simulation (Fig. 2c).

Having established the models were identifiable and parameters recoverable, we performed Bayesian model selection on the data from our participants. Participant's choices were best characterised by the $3\alpha1\beta$ model. This indicated that the learning process underlying the choices is most accurately captured by assuming separate learning rates for each recipient ($\alpha_{self}$, $\alpha_{other}$

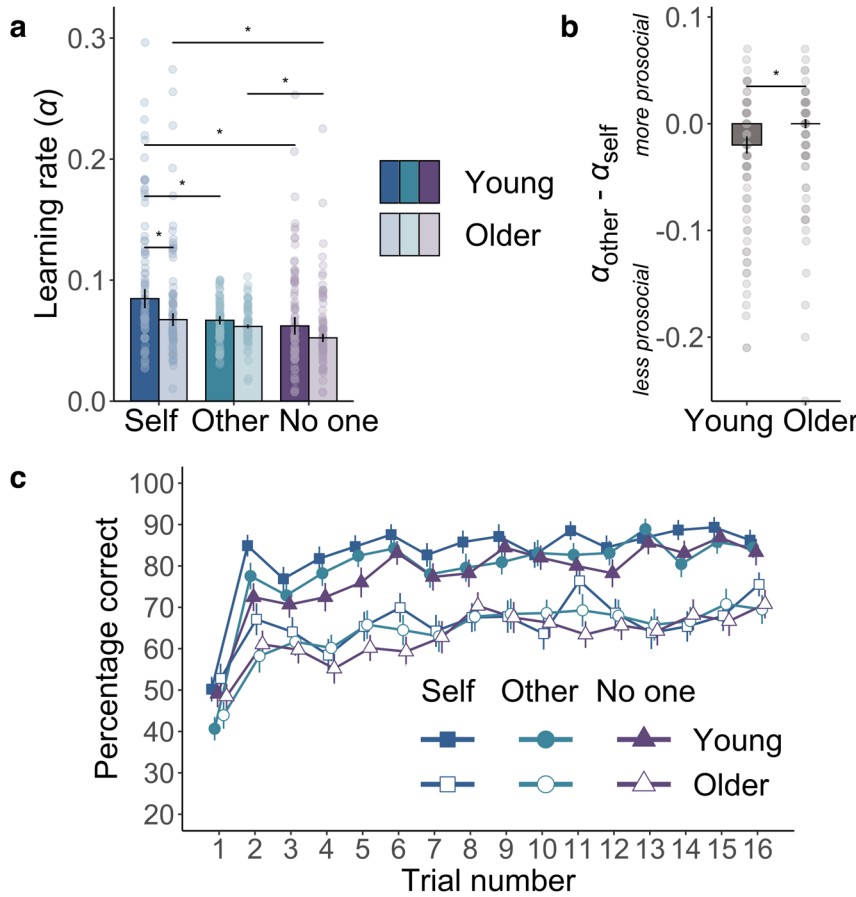

**Fig. 3 Age-group differences in accuracy and learning rates. a** Comparison of learning rates ($\alpha$) from the computational model shows older adult's prosocial learning was preserved. Learning rates were calculated in the self (blue), other (green) and no one (purple) conditions, and are plotted separately for the young (strong colours) and older (pastel colours) age groups. Older adults' learning rates in the other condition did not differ from the self condition ($V = 1150$, $Z = -1.45$, $r_{(75)} = 0.17$ [0.01, 0.38], $P = 0.15$, $BF_{01} = 1.08$) or from young adults' prosocial learning rates ($W = 3042$, $Z = -0.86$, $r_{(150)} = 0.07$ [0.003, 0.24], $P = 0.39$, $BF_{01} = 4.26$). In contrast, young participants learned faster for themselves than another person ($V = 659$, $Z = -4.04$, $r_{(75)} = 0.47$ [0.26, 0.63], $P < 0.001$), $n = 150$ (75 young, 75 older). Bars show group median, error bars are standard error of the median, asterisks represent significant two-sided between-group and within-group Wilcoxon $t$ tests ($P < 0.05$). **b** Median difference between $\alpha_{other}$ and $\alpha_{self}$ illustrates the significant age group * recipient [self vs. other] interaction ($b = 0.02$ [0.002, 0.03], $Z = 2.29$, $P = 0.02$), $n = 150$ (75 young, 75 older). Error bars are standard error of the median, the asterisk represents the significant interaction from the robust linear mixed-effects model. **c** Group-level learning curves showing choice behaviour in the three recipient conditions (self: blue squares, other: green circles, no one: purple triangles) for each age group (young: filled, older: empty shapes). Trials are averaged over the three blocks (48 trials total per recipient presented in three blocks of 16 trials) for the self, other and no one recipients, $n = 152$ (75 young, 77 older). Points show group mean, error bars are standard error of the mean. Source data are provided as a Source Data file.

and $\alpha_{no\ one}$). This model fit the data best (exceedance probability = 97%; $\Delta BIC_{int} = 122$; Fig. 2d) and predicted choices well ($R^2 = 51\%$; see 'Methods' for further details).

**Higher learning rates are associated with better performance.** As a final check of the robustness of our model and to enable clear interpretation of any differences in learning rate between recipients and age groups (see ref. [52]), we conducted an additional simulation experiment. We simulated data from 10,000 participants using the $3\alpha1\beta$ model. This created 30,000 values of $\alpha$ from the three recipient conditions, spanning the full range of possible values from 0 to 1 (see 'Methods'). For each, we quantified the associated performance as the percentage of times the synthetic participant chose the high probability of reward option, averaged across the blocks for the relevant recipient. Plotting the learning rates against performance (Fig. 2e) shows that the optimal value of $\alpha$ is ~0.55. This is higher than all the values of $\alpha$ found on our task in any recipient condition for either age group. Therefore, higher learning rates were associated with better performance. We further established this link by correlating learning rates and

performance in the empirical data from our participants. We found a strong correlation between learning rates and performance overall ($r_{s(150)}$ [95% confidence interval] $= 0.58$ [0.46, 0.68], $P < 0.001$; Fig. 2f) and in each recipient and age-group combination (Supplementary Table 3).

**Older adults are slower at learning for themselves, but prosocial learning is preserved.** Next, we used this validated computational model to test our hypotheses as to whether there were group differences in learning rates when learning to reward self, other or no one. Two participants had learning rates for two of the three recipients more than three standard deviations (SDs) above the mean ($\alpha_{self}$ 6.68 and $\alpha_{no\ one}$ 9.64; $\alpha_{self}$ 7.96 and $\alpha_{other}$ 3.78 SDs above the mean) and were excluded from all analysis of learning rates. We analysed the condition-specific learning rates from our best-fitting computational model using a robust linear mixed-effects model (RLMM; see 'Methods'). The RLMM fixed effects were age group (young, older), recipient (self, other and no one), as well as the age group * recipient interaction. While this RLMM includes all three recipient conditions, it generates coefficients (main effect and the

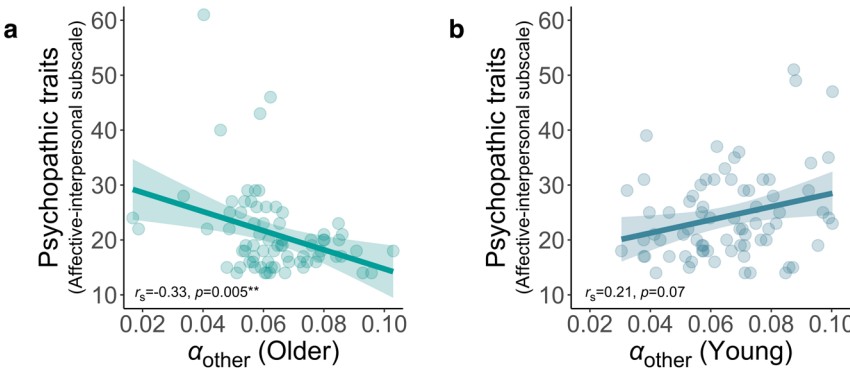

**Fig. 4 Correlations between prosocial learning rates ($\alpha_{\text{other}}$) and scores on the affective-interpersonal subscale of the Self-Report Psychopathy Scale.** **a** For older adults, levels of subclinical self-reported psychopathic traits are negatively correlated with prosocial learning rates ($r_{s(74)} = -0.33$ [$-0.52$, $-0.11$], $P = 0.005$, false discovery rate (FDR) corrected $P = 0.03$). **b** There is no significant relationship for young adults ($r_{s(74)} = 0.21$ [$-0.02$, $0.42$], $P = 0.07$, FDR-corrected $P = 0.22$) and the correlation is significantly more negative for older than young adults ($Z$ test of the difference between correlations $Z = 3.28$, $P = 0.001$). Relative prosocial learning ($\alpha_{\text{other}} - \alpha_{\text{self}}$) is also negatively correlated with self-reported psychopathic traits in older adults ($r_{s(74)} = -0.25$ [$-0.45$, $-0.02$], $P = 0.03$) but not younger adults ($r_{s(74)} = 0.11$ [$-0.12$, $0.33$], $P = 0.34$; difference $Z = 2.18$, $P = 0.03$). Age-group differences in self-reported psychopathic traits and the correlation between $\alpha_{\text{other}}$ and self-reported psychopathic traits, for older adults only, also remained significant when excluding extreme scores (>3 SDs from the mean) on the psychopathic traits self-report measure (Supplementary Tables 6 and 7). Shaded areas represent 95% confidence intervals. Source data are provided as a Source Data file.

interaction with age group) contrasting pairs of recipient conditions —[self vs. other] and [self vs. no one]. These are more interpretable than an omnibus test that would not show which recipient conditions were driving an effect or interaction. We followed up these results with planned comparisons, between the older and younger group in each recipient condition, and between pairs of recipient conditions within each age group.

Across age groups, participants showed a higher learning rate when rewards were for themselves, compared to for another person (recipient [self vs. other]: $b = -0.02$ [$-0.03$, $-0.01$], $Z = -4.79$, $P < 0.001$). Importantly, however, this pattern differed between age groups. The difference between self-relevant and prosocial learning rates was reduced in older compared to younger adults (recipient [self vs. other] * age-group interaction: $b = 0.02$ [0.002, 0.03], $Z = 2.29$, $P = 0.02$). Between-group comparisons showed older adults learnt more slowly for themselves compared to younger adults ($W = 3512$, $Z = -2.63$, $r_{(150)} = 0.22$ [0.06, 0.36], $P = 0.009$). However, prosocial learning was preserved, with a Bayes factor suggesting strong evidence of no difference in $\alpha_{\text{other}}$ between young and older adults ($W = 3042$, $Z = -0.86$, $r_{(150)} = 0.07$ [0.003, 0.24], $P = 0.39$, $BF_{01} = 4.26$). Within-subject comparisons of $\alpha_{\text{self}}$ and $\alpha_{\text{other}}$ in each age group showed that young adults had higher learning rates for themselves, relative to another person ($V = 659$, $Z = -4.04$, $r_{(75)} = 0.47$ [0.26, 0.63], $P < 0.001$). In contrast, older adults showed no significant difference between learning rates for self and other ($V = 1150$, $Z = -1.45$, $r_{(75)} = 0.17$ [0.01, 0.38], $P = 0.15$, $BF_{01} = 1.08$).

As expected, across age groups learning was slower for no one than self (recipient [self vs. no one]: $b = -0.02$ [$-0.03$, $-0.01$], $Z = -4.57$, $P < 0.001$). Unlike $\alpha_{\text{self}}$ vs. $\alpha_{\text{other}}$, learning for self compared to no one did not interact with age group (recipient [self vs. no one] * age-group interaction: $b = 0.008$ [$-0.006$, 0.02], $Z = 1.15$, $P = 0.25$). Within-subject comparisons between $\alpha_{\text{self}}$ and $\alpha_{\text{no one}}$ in each age group showed that both groups learnt preferentially for themselves compared to no one (young adults: $V = 928$, $Z = -2.62$, $r_{(75)} = 0.30$ [0.07, 0.51], $P = 0.009$; older adults $V = 901$, $Z = -2.76$, $r_{(75)} = 0.32$ [0.09, 0.53], $P = 0.006$). There was no significant difference between the age groups in $\alpha_{\text{no one}}$ but also no evidence in support of the null ($W = 3241$, $Z = -1.61$, $r_{(150)} = 0.13$ [0.01, 0.29], $P = 0.11$, $BF_{01} = 2.04$).

Considering differences in learning between $\alpha_{\text{other}}$ and $\alpha_{\text{no one}}$, young adults did not differentiate between another person and no one, with strong Bayesian evidence for no difference ($V = 1533$, $Z = -0.57$, $r_{(75)} = 0.07$ [0.003, 0.31], $P = 0.57$, $BF_{01} = 5.08$). In contrast, older adults had higher learning rates for another person, compared to no one ($V = 976$, $Z = -2.37$, $r_{(75)} = 0.27$ [0.05, 0.49], $P = 0.02$). Crucially, this shows that older adults' lack of differentiation between self and other was not simply because they were insensitive to the recipient condition.

Finally, we also observed an effect of age on both learning rates overall and temperature parameters. Older adults showed slower learning overall compared to younger adults ($b = -0.02$ [$-0.03$, $-0.01$], $Z = -3.73$, $P < 0.001$) and higher levels of exploration of choice options (median $\beta$ young: 0.05, older: 0.19, $W = 1511$, $Z = -4.89$, $r_{(150)} = 0.40$ [0.26, 0.53], $P < 0.001$; Supplementary Fig. 1).

In summary, older adults prosocial learning was preserved at the same rate as young adults, despite age-related declines in self-relevant learning rates. In other words, young adults learned faster for themselves than others, but older adults distinguished between themselves and others significantly less than the young participants. Only older adults, not young adults, distinguished between rewards for another person and no one.

**Participants perform better for themselves, compared to no one.** For completeness, we also tested the effects of recipient and age group on trial-by-trial tendency to pick the high-reward stimuli (Fig. 3c). In addition to the main effect of trial number ($b = 1.70$ [1.29, 2.13], $Z = 7.97$, $P < 0.001$), showing learning, these models revealed older adults chose the high-reward option less frequently (mean for young: 80%, older: 64%, $b = -1.20$ [$-1.65$, $-0.70$], $Z = -4.84$, $P < 0.001$), and improved less during the task (trial number * age-group interaction $b = -0.81$ [$-1.34$, $-0.27$], $Z = -2.95$, $P = 0.003$) across recipients. When averaging across age groups, performance was better for the self (75%), compared to no one (70%; $b = -0.36$ [$-0.68$, $-0.05$], $Z = -2.28$, $P = 0.02$). However, there was not a significant difference between accuracy for other (72%) and self ($b = -0.22$ [$-0.49$, 0.05], $Z = -1.63$, $P = 0.10$), or any significant interactions between the age group and the recipient ($bs < 0.16$, $Zs < 0.96$, $Ps > 0.34$).

**Self-reported psychopathic traits are lower and associated with prosocial learning in older adults**. Finally, we examined individual variability in self-reported psychopathic traits, considering age-related differences and influence on prosocial learning. Several studies have suggested that individual differences in subclinical self-reported psychopathic traits can be meaningfully and accurately captured in community samples and often parallel findings in criminal offenders[45]. Critically, self-reported psychopathic traits are closely linked to alterations in social behaviour and willingness to help others. Therefore, we also asked participants to complete the Self-Report Psychopathy Scale (SRP-IV-SF)[40]. The SRP is a self-report measure of psychopathic traits in community samples that assesses traits with the same factor structure as in measures used for clinical psychopathies, such as antisociality and interpersonal effect. The measure robustly captures the latent structure of clinical psychopathy to ensure parallels can be drawn between normal individual differences in the community and clinical samples (see 'Methods'). One participant in each age group had missing questionnaire data and are not included in these analyses. Self-reported psychopathic traits are consistently divided into two components that this scale measures: core 'affective-interpersonal' traits, which capture lack of empathy and guilt; and 'lifestyle-antisocial' traits, which capture impulsivity and antisocial tendencies. Comparing the two age groups on these scales showed that older participants had significantly lower scores than young participants on both the core affective-interpersonal (young mean = 24.36, older mean = 21.09, $W = 3558$, $Z = -3.15$, $r_{(148)} = 0.26$ [0.11, 0.40], $P = 0.002$) and the lifestyle-antisocial subscales (young mean = 22.89, older mean = 20.27, $W = 3471$, $Z = -2.82$, $r_{(148)} = 0.23$ [0.09, 0.38], $P = 0.005$). These findings suggest that both components of self-reported psychopathic traits were reduced in older, compared to younger, adults.

Next, we sought to test our hypothesis that individual differences in self-reported core psychopathic traits would explain variability in learning rates, specifically for prosocial learning. We observed a significant negative relationship between $\alpha_{\text{other}}$ and self-reported core psychopathic traits among older participants ($r_{s(74)} = -0.33$ [-0.52, -0.11], $P = 0.005$; Fig. 4a). Intriguingly, this relationship was significantly more negative ($Z = 3.28$,

$P = 0.001$) than the equivalent correlation in young adults, which was not significant ($r_{s(74)} = 0.21$ [-0.02, 0.42], $P = 0.07$; Fig. 4b). This pattern of results was the same when correlating relative prosocial learning rate ($\alpha_{\text{other}} - \alpha_{\text{self}}$) with self-reported psychopathic traits. We also conducted control analyses, correlating the same pairs of variables but using partial correlations controlling for $\beta$. The negative relationship between prosocial learning (when quantified as $\alpha_{\text{other}}$ or $\alpha_{\text{other}} - \alpha_{\text{self}}$) and self-reported psychopathic traits was still present for older adults, showing that the correlations with learning rates were independent of individual choice exploration (all $Ps < 0.05$; see Supplementary Table 4). The negative relationship between $\alpha_{\text{other}}$ and self-reported psychopathic traits for older adults also remained significant after applying false discovery rate (FDR) correction for multiple comparisons across this correlation and the five other age group and recipient combinations (see Supplementary Table 5). Moreover, the finding that no significant correlations were apparent between self-reported psychopathic traits and $\alpha_{\text{self}}$ or $\alpha_{\text{no one}}$ ($Ps > 0.15$; Supplementary Table 5), suggests a specificity in the relevance of psychopathic traits to prosocial learning in older adults.

Given the age-group differences in levels of self-reported psychopathic traits and their correlations with prosocial learning, our final analysis considered possible indirect effects of age group on relative prosocial learning rate ($\alpha_{\text{other}} - \alpha_{\text{self}}$), through scores on the core affective-interpersonal subscale of the psychopathic traits measure. A standard mediation model (Fig. 5a) did not show evidence for a significant indirect effect. However, as would be predicted if the link between self-reported psychopathic traits and prosocial learning depends on age, there was a significant indirect effect in a moderated mediation model. This revealed an indirect effect of age group on relative prosocial learning rates via self-reported psychopathic traits for older adults (unstandardised indirect effect = 0.006 [0.001, 0.01], $P = 0.006$, proportion mediated = 0.30) but not for young adults (unstandardised indirect effect = -0.001 [-0.007, 0.006], $P = 0.67$; see Fig. 5b for standardised coefficients). In summary, young and older adults differed in levels of self-reported psychopathic traits and whether or not self-reported psychopathic trait scores were associated with their prosocial learning rates.

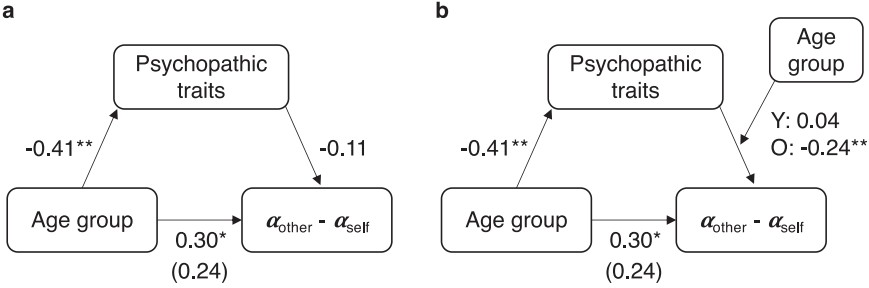

**Fig. 5 Indirect effect of age group on relative prosocial learning ($\alpha_{\text{other}} - \alpha_{\text{self}}$) via self-reported psychopathic traits for older adults only. a** A standard mediation model does not show evidence of an indirect effect of age group on relative prosocial learning via self-reported affective-interpersonal psychopathic traits. Although accounting for self-reported psychopathic traits means the significant direct effect of age group on relative prosocial learning (standardised coefficient = 0.30 [0.02, 0.57], $t_{(146)} = 2.14$, $P = 0.03$) becomes non-significant (standardised coefficient = 0.24 [-0.04, 0.53], $t_{(145)} = 1.72$, $P = 0.09$), self-reported psychopathic traits do not predict relative prosocial learning overall so there is no indirect effect. **b** Statistical evidence of a moderated mediation is revealed when accounting for differences between young and older adults in how self-reported psychopathic traits predict relative prosocial learning. An indirect path via self-reported psychopathic traits exists for older but not younger adults. Older adults have lower levels of self-reported psychopathic traits (standardised coefficient = -0.41 [-0.66, -0.16], $t_{(146)} = -3.23$, $P = 0.002$) and lower self-reported psychopathic traits are associated with increased relative prosocial learning for older (standardised coefficient = -0.24 [-0.41, -0.08], $t_{(72)} = -2.97$, $P = 0.004$) but not younger (standardised coefficient = 0.04 [-0.19, 0.27], $t_{(72)} = 0.34$, $P = 0.74$) adults. Accounting for self-reported psychopathic traits and the interaction with age group means the significant effect of age group on relative prosocial learning becomes non-significant (standardised coefficient = 0.24 [-0.04, 0.51], $t_{(144)} = 1.71$, $P = 0.09$). Self-reported psychopathic traits are scores on the affective-interpersonal subscale of the Self-Report Psychopathy scale, Y: young, O: older, $n = 150$ (75 young, 75 older), values are standardised coefficients from robust linear models, asterisks represent significant effects (*$P <$ 0.05, **$P <$ 0.01). Source data are provided as a Source Data file.

## Discussion

Reinforcement learning is a fundamental process for adaptive behaviour in many species. However, existing studies have largely focused on young people and self-relevant learning, but the decisions we make often occur in a social context[53] and our actions affect outcomes for others. Here, we applied computational models of reinforcement learning to the question of ageing-related changes in self-relevant and prosocial learning. We found a clear decrease in learning rates for self-relevant rewards in older, compared to younger, adults. Intriguingly, despite this reduction in self-relevant learning, learning rates for outcomes that affected others did not differ between older and young adults, with Bayesian analyses supporting no difference. Moreover, not only did older adults have preserved prosocial learning rates, but age was also associated with lower self-reported psychopathic traits, which were specifically linked to prosocial learning in older adults.

Models of learning are a powerful tool for understanding prosocial behaviour. By isolating the learning rate, we can precisely examine the influence of reward history on learning. We robustly replicated previous findings that self-relevant learning can be computationally separated from prosocial learning[7], with different learning rates for different recipients providing the best explanation of behaviour. Including separate learning rates improved the model fit and, on average across participants, learning rates were higher for self-relevant rewards, compared to when someone else received the reward. However, this increase in learning rates for self, compared to other, was reduced in older adults who showed higher prosocial learning rates, relative to their own self-relevant learning rates, than young adults. As expected, older adults learnt more slowly for themselves than young adults but in the prosocial condition, the learning rates did not significantly differ between the age groups. Bayesian analyses additionally confirmed that prosocial learning was preserved between young and older adults.

As with much research on age-related changes in cognitive and social tasks, our key finding that older adults are relatively more prosocial on a learning task could be interpreted as due to changes in ability, motivation, or a combination of these factors. Importantly, learning rates were not associated with executive function or an age-standardised measure of intelligence. We also show that our results remain the same after controlling for these measures. These findings suggest that the observed decline in self-relevant learning rates, but the preservation of prosocial learning rates, for older adults are not explained by changes in these broad abilities. Considering learning more specifically, a recent comparison of motivation and ability as explanations for ageing-related reductions in model-based strategies during self-relevant learning supported the limited cognitive abilities account[54]. Our finding that learning rates for self-relevant outcomes were reduced in older adults is in line with alterations in the neuro-cognitive systems required for successful learning. Research combining models of learning with neuroimaging and pharmacological manipulations suggests ageing reduces the ability to generate reward prediction errors[18] (but see refs. [55,56]) due to declines in dopamine functioning[28,57] (also see refs. [58,59]). Differences in motivation could also be applicable for self-relevant learning as the subjective value of financial outcomes is also likely to decrease in older age, due to changes in wealth across the lifespan[35].

Our findings suggest that despite declines in learning ability associated with ageing, prosocial learning—learning to help others—is preserved. This finding aligns with emerging literature showing older adults may be more prosocial than younger adults[30,31,60,61]. The possibility that relatively preserved prosocial learning is related to increased prosocial motivation is in line with

our observed link between learning rates and psychopathic traits. Self-reported psychopathic traits were significantly reduced in our older adult sample, dovetailing with similar previous findings on this trait[46,47] and broader trait benevolence[34]. Importantly, we found self-reported psychopathic traits in older adults negatively correlated with prosocial learning rates. The tendency to learn faster for oneself compared to another person was most reduced, and even reversed, in the older people lowest on psychopathic traits. Notably, this negative correlation between self-reported psychopathic traits and prosocial learning was only found for older adults. This suggests that age-related differences in prosocial learning could be linked to basic shifts in individual traits and motivations over the lifespan, not just to domain-general reductions in cognitive abilities. The idea that social motivations become more influential in learning and decision-making with age has also been suggested based on studies of social rewards such as smiling faces or hypothetical time spent with social partners[62–64]. Taken together, this work suggests that strategies to support healthy ageing might benefit from leveraging potentially preserved social motivations, a hypothesis future research could address.

Importantly, our task included a control condition where no one benefitted. This was essential to establish that the lack of difference between self and other learning rates for older adults was not simply due to an age-related reduction in the absolute dynamic range as maximum learning rates decrease. Older adults had higher learning rates for both others and themselves, compared to this control condition. In contrast, young adults did not differentiate another person from no one. Older adults, therefore, showed a relative increase in learning rates that was specific to the prosocial condition. This is also evidence against the idea that lower learning rates in older people are reflective of a general reduced sensitivity to who gets the reward. It is interesting to note that the magnitude of the decrease in self-relevant learning rates associated with being older, compared to younger, is similar to the decrease shown by the young participants when learning for someone else, compared to themselves. The preservation of prosocial learning rates between age groups may seem at odds with the decreased self-relevant learning rates in our sample of older adults and existing evidence of underlying neurobiological changes. However, learning from outcomes for self and other have been linked to distinct regions of the brain, shown through human neuroimaging[7,65,66], and causally in monkeys with focal lesions[67].

Taken together, our results add to a growing body of literature suggesting age-related increases in prosocial motivation[30,31,60,61]. If this is the case, the next question is how and why this happens, as there are many possible reasons to be prosocial. For example, prosocial behaviours can be motivated by reputational concerns[68], the 'warm glow' of helping[69–71], or vicarious reinforcement from positive outcomes for others[72]. In our procedure, we were very careful to prevent reputational concerns influencing learning to help others. Participants underwent an extensive procedure to introduce them to another participant but to hide information about their age and identity. This meant we could assess the tendency for prosocial learning in a situation where reputational motivations were excluded, and identity-based influences were controlled. Using a reinforcement-learning task, in which performance generates positive outcomes for others, also focuses on vicarious rewards from the outcome, rather than the warm glow associated with the action of helping. Thus, the increase in prosocial learning rates, relative to self-relevant learning rates, suggests older adults are reinforced by outcomes for others and themselves more similarly than younger adults. Many prosocial measures such as the dictator game[73] are costly, requiring direct trade-offs between outcomes for oneself and others. This also makes it hard to detect whether changes are in

the value of outcomes for oneself, or outcomes for others, or both. In contrast, separating self-relevant learning, prosocial learning, and the control condition allows us to differentiate increases in the value of prosocial outcomes from decreases in the value of outcomes for the self. Our results are consistent with older adults having both decreased self-relevant learning rates (compared to young adults) and increased prosocial learning rates (compared to their performance in the control condition).

Our results also support the advantages of a model-based approach for understanding both prosocial behaviour and ageing. The model-based parameters were more sensitive to the effects of interest than general measures of performance on the task. The model comparison process is able to provide important information about how the learning process takes place, which cannot be revealed from performance measures alone. We showed that learning was best represented by a separate learning rate for each recipient. Moreover, this approach can capture additional latent parameters that drive behaviour, such as the inverse temperature, which indexes how closely participants follow the stimulus value. We demonstrate that a single inverse-temperature parameter best explained behaviour during learning across recipient conditions, despite learning rates being distinct.

While our procedure and task have many benefits, it is important to also recognise limitations. To test for age-related differences in prosocial learning, we recruited a group of older adults and a group of younger adults. This increases the power to detect differences, but we are unable to assess at what age or how quickly changes take place. Also, while our age groups were matched on years of education and IQ test performance, the recruitment from university databases or issues around self-selection may mean that the levels of education and IQ test performance in our sample are not completely representative of the general population. Further studies could include samples recruited entirely from the community and participants across the whole adult lifespan. Future studies should also assess the timescale after which older participants reach similar ceiling levels of performance as younger adults, or whether they are never able to reach the same level of performance. Of similar importance is the question of whether older adults may be able to sustain higher levels of motivation over an extended period of time compared to younger adults, possibly compensating for slower learning speeds. In this study, due to time constraints and the presence of several conditions, we are only able to derive conclusions from a limited time window. Moreover, previous research has suggested that individual differences in empathy—the ability to vicariously experience and understand others' affect—might relate to differences in prosocial learning[7]. Empathy is positively associated with affective-interpersonal psychopathic traits and might also relate to motivation to help others[74,75]. Further studies could also assess how empathy predicts changes in prosocial learning across the lifespan. Finally, future studies could manipulate the identity of the recipient as we show preservation in the tendency to help others when there are no particular characteristics known about the other person, but these effects might additionally be modulated by factors such as perceived social distance.

To conclude, we find evidence that despite declines in self-relevant learning in older adults, the ability to learn which actions benefit others is preserved. Moreover, the bias with which people favour self-relevant outcomes is reduced. Not only do older adults have relatively preserved prosocial learning, they also report lower levels of core psychopathic traits that index lack of empathy and guilt, and these traits are linked to variability in prosocial learning. These findings could have important implications for our understanding of reinforcement learning and theoretical accounts of healthy ageing.

## Methods

**Participants**. We recruited 80 young participants and 80 older participants using the same recruitment methods in order to match the samples as closely as possible. Participants were recruited from university databases, which included students and members of the community, social media, and adverts in local newspapers. We excluded anyone who was currently studying or had previously studied psychology, and no one took part for course credit. Additional exclusion criteria were previous or current neurological or psychiatric disorder, non-normal or non-corrected to normal vision and, for the older sample, scores on the Addenbrooke's Cognitive Examination that indicate potential dementia (cut-off score 82)[48]. This sample size gave us 88% power to detect a medium-size effect ($d = 0.5$).

Five young and three older participants were excluded due to: diagnosis of a psychiatric disorder at the time of testing (one young participant); previous study of psychology (two young participants); and incomplete or low-quality data (two young and three older participants). This left a final sample of 152 participants, 75 young adults (44 females aged 18–36, mean = 23.07) and 77 older adults (40 females aged 60–80, mean = 69.84). Two older participants were excluded from all analyses involving learning rates due to each having two learning rate estimates more than three standard deviations (SDs) above the mean (for one $\alpha_{self}$ was 6.68 SDs above the mean and $\alpha_{no}$ one 9.64 SDs above the mean; for the second $\alpha_{self}$ was 7.96 and $\alpha_{other}$ 3.78 SDs above the mean). One further participant from each age group was missing data on the SRP measure so are not included in the relevant analyses.

Participants were paid at a rate of £10 per hour plus an additional payment of up to £5 depending on the number of points they earned for themselves during the task. They were also told the number of points that they earned in the prosocial condition would translate into an additional payment of up to £5 for the other participant (see details of the task below). All participants provided written informed consent and the study was approved by the Oxford University Medical Sciences Inter Divisional Research Ethics Committee and National Health Service Ethics.

**Prosocial learning task**. The prosocial learning task is a probabilistic reinforcement-learning task, with rewards in one of three recipient conditions: for the participant themselves (self), for another participant (other; prosocial condition), and for neither person (no one; control condition)[7]. Each trial presents two symbols, one associated with a high (75%) probability of gaining points and the other with a low (25%) probability of gaining points. The two symbols were randomly assigned to the left or right side of the screen and selected via a corresponding button press. Participants select a symbol then receive feedback on whether they obtained points or not (see Fig. 1b) so learn over time which symbol maximises rewards. The experiment was subdivided into blocks, i.e., 16 trials pairing the same two symbols for the same recipient. Participants completed three blocks, a total of 48 trials, in each recipient condition, resulting in 144 trials overall (see Supplementary Information for trial structure). Blocks for different recipients were pseudo-randomly ordered such that the same recipient block did not occur twice in a row.

On trials in the self condition, points translated into increased payment for the participant themselves. These blocks started with 'play for you' displayed and had the word 'you' at the top of each screen. Blocks in the no one condition had 'no one' in place of 'you' and points were not converted into any extra payment for anyone. In the prosocial other condition, participants earned points that translated into additional payment for a second participant, actually a confederate. Participants were told that this payment would be given anonymously, they would never meet the other person, and that the person was not even aware of them completing this task (see Supplementary Information). The name of the confederate, gender-matched to the participant, was displayed on these blocks at the start and on each screen (Fig. 1b). Thus, participants were explicitly aware of how their decisions affected on each trial. Stimuli were presented using Presentation v17 (Neurobehavioral Systems— https://www.neurobs.com/).

## Questionnaire measures

*Dementia screening and executive function.* Older adults were screened for dementia using the Addenbrooke's Cognitive Examination (ACE-III)[48]. The ACE examines five cognitive domains; attention, memory, language, fluency and visuospatial abilities. The ACE-III is scored out of 100 and as a screening tool, a cut-off score of 82/100 denotes significant cognitive impairment. We also used scores on the attention and memory domains in control analyses as proxies for executive function in older adults.

*General intelligence.* All participants completed the Wechsler Test of Adult Reading (WTAR)[49] as a measure of IQ test performance. The WTAR requires participants to pronounce 50 words that have an unusual grapheme-to-phoneme translation. This means the test measures reading recognition or existing knowledge of the words, rather than the ability to apply rules for pronunciation. The WTAR was developed and standardised with the Wechsler Memory and Adult Intelligence Scales and correlates highly with these measures[76]. Standardisation involves adjusting for healthy age-related differences. The test is suitable for participants aged 16–89, covering our full sample, and scores in older age have been shown to correlate with cognitive ability earlier in life[77].

*Psychopathic traits.* Participants completed the short form of the Self-Report Psychopathy Scale 4$^{th}$ Edition (SRP-IV-SF)[40]. This scale consists of 29 items, 7 each measuring: interpersonal, affective, lifestyle and antisocial tendencies (plus 'I have been convicted of a serious crime'). We used the two-factor grouping, summing the core, affective-interpersonal items and separately, the lifestyle-antisocial items for use in the analysis. The robust psychometric properties of this measure have been established in community[78] and offender populations through construct and convergent validity[79], internal consistency and reliability[40].

### Procedure

*Role assignment.* To enhance the belief that points earned in the prosocial condition benefitted another person, we conducted a role-assignment procedure based on a set-up used in several studies of social decision-making[80,81]. Participants were instructed not to speak and wore a glove to hide their identity. A second experimenter brought the confederate, also wearing a glove, to the other side of the door. Participants only ever saw the gloved hand of the confederate, but they waved to each other to make it clear there was another person there (Fig. 1a). The experimenter tossed a coin to determine who picked a ball from the box first and then told the participants which roles they had been assigned to, based on the ball they picked. Our procedure ensured that participants always ended up in the role of the person performing the prosocial learning experiment. Participants were unaware of the age of the other person, but the experimenter used a name for them, suggesting their gender was the same as the participant.

*Task procedure.* Participants received instructions for the learning task and how the points they earned would be converted into money for themselves and for the other participant. Instructions included that the two symbols were different in how likely it was that choosing them lead to points but that which side they appeared on the screen was irrelevant. Participants then completed one block of practice trials before the main task and were aware outcomes during the practice did not affect payment for anyone. After the task, participants completed the self-report measure of psychopathic traits and the dementia screening.

### Computational modelling.

We modelled learning during the task with a reinforcement-learning algorithm[16], creating variations of the models through the number of parameters used to explain the learning rate and temperature parameters in the task[82]. The basis of the reinforcement-learning algorithm is the expectation that an action (or stimulus) $a$ will provide reward on the following trial. This expected value, $Q_{t+1}(a)$ is quantified as a function of current expectations $Q_t(a)$ and the prediction error $\delta_t$, which is scaled by the learning rate $\alpha$:

$$Q_{t+1}(a) = Q_t(a) + \alpha \times \underbrace{[r_t - Q_t(a)]}_{\text{Prediction error } \delta_t} \quad (1)$$

Where $\delta_t$, the prediction error, is the difference between the actual reward experienced on the current trial $r_t$ (1 for reward and 0 for no reward) minus the expected reward on the current trial $Q_t(a)$.

The learning rate $\alpha$ therefore determines the influence of the prediction error. A low learning rate means new information affects expected value to a lesser extent. The softmax link function quantifies the relationship between the expected value of the action $Q_t(a)$ and the probability of choosing that action on trial $t$:

$$p_t[(a|Q_t(a))] = \frac{e^{(Q_t(a)/\beta)}}{\sum_{a'} e^{(Q_t(a')/\beta)}} \quad (2)$$

The temperature parameter $\beta$ represents the noisiness of decisions—whether the participant explores or always chooses the option with the highest expected value. A high value for $\beta$ means choices seems random as they are equally likely irrespective of the expected value of each option. A low $\beta$ leads to choosing the option with the greatest expected value on all trials.

### Model fitting.

We used MATLAB 2019b (The MathWorks Inc) for all model fitting and comparison. To fit the variations of the learning model (see below) to (real and simulated) participant data, we used an iterative maximum a posteriori (MAP) approach[50,51]. This approach involves implementing two levels: the lower level of the individual subjects and the higher level reflecting our full sample. An MAP approach provides a better estimation than a single-step maximum likelihood estimation (MLE) alone by being less susceptible to the influence of outliers. For the real participant data, we fit the model across groups to provide the most conservative comparison, so this full sample combined young and older participants.

For the MAP procedure, we initialised group-level Gaussians as uninformative priors with means of 0.1 (plus some added noise) and variance of 100. During the expectation, we estimated the model parameters ($\alpha$ and $\beta$) for each participant using an MLE approach, calculating the log-likelihood of the subject's series of choices given the model. We then computed the maximum posterior probability estimate, given the observed choices and given the prior computed from the group-level Gaussian, and recomputed the Gaussian distribution over parameters during the maximisation step. We repeated expectation and maximisation steps iteratively until convergence of the posterior likelihood summed over the group or a maximum of 800 steps. Convergence was defined as a change in posterior

likelihood <0.001 from one iteration to the next. Note that bounded free parameters were transformed from the Gaussian space into the native model space via appropriate link functions (e.g., a sigmoid function in the case of the learning rates) to ensure accurate parameter estimation near the bounds. The detailed code for the models and implementation of the fitting algorithm[83] can be found at https://doi.org/10.17605/osf.io/xgw7h.

### Model comparison.

Our hypotheses generated four models to compare that differed in whether the model parameters ($\alpha$ and $\beta$) for each participant had one value across recipient conditions or depended on the recipient (self, other and no one). For model comparison, we calculated the Laplace approximation of the log model evidence (more positive values indicating better model fit[84]) and submitted these to a random-effects analysis using the spm_BMS routine[85] from SPM 8 (http://www.fil.ion.ucl.ac.uk/spm/software/spm8/). This generates the exceedance probability: the posterior probability that each model is the most likely of the model set in the population (higher is better, over 0.95 indicates strong evidence in favour of a model). For the models of real participant data, we also calculated the integrated BIC (lower is better[50,51]) and R2 as additional measures of model fit. To calculate the model R2, we extracted the choice probabilities generated for each participant on each trial from the winning model. We then took the squared median choice probability across participants. The $3\alpha1\beta$ model had the best evidence on all measures (see Supplementary Table 2).

### Simulation experiments.

We used simulation experiments to assess that our experiment allowed us to dissociate models of interest, as well as parameters of interest within the winning model. We simulated data from all four models to establish that our model comparison procedure (see above) could accurately identify the best model across a wide range of parameter values. For this model identifiability analysis, we simulated data from 150 participants, drawing parameters from distributions commonly used in the reinforcement-learning literature[86,87]. Learning rates ($\alpha$) were drawn from a beta distribution (betapdf (parameter,1.1,1.1)) and softmax temperature parameters ($\beta$) from a gamma distribution (gampdf(parameter,1.2,5)). We fitted the models to this simulated data set using the same MAP process as applied to the experimental participants' data and repeated this whole procedure ten times. By plotting the confusion matrices of average exceedance probability (across the ten runs; Fig. 2a) and how many times each model won (Fig. 2b), we show the models are identifiable using our model comparison process.

Our winning model contained four free parameters ($\alpha_{self}$, $\alpha_{other}$, $\alpha_{no\ one}$, $\beta$). To assess the reliability of this model and the interpretability of the free parameters, we also performed parameter recovery on simulated data (see Supplementary Information for procedure) as recommended for modelling analyses that use a 'data first' approach[82,88]. We simulated choices 1296 times using our experimental schedule and fitted them using MAP. We found strong Pearson's correlations between the true simulated and fitted parameter values (all $rs > 0.7$, see Fig. 2c), suggesting our experiment was well suited to estimate the model's parameters.

Finally, we conducted a principled simulation experiment to identify the optimal learning rate in our task and examine the link between learning rates and performance. We simulated data from 10,000 participants with learning rates ($\alpha$) and softmax temperature parameters ($\beta$) drawn from beta and gamma distributions, respectively, as described above. We bounded the $\beta$ parameters at 0 and 0.3 to reflect the range shown by our participants. In line with our winning model, simulated participants had a separate learning rate for each recipient condition, and these spanned the full range of possible $\alpha$ values from 0 to 1. This generated 30,000 learning rate values (10,000 participants and three conditions). For each, we quantified performance as the proportion of times the simulated participant chose the 'correct' high-reward option in the relevant condition. Results showed that average performance improved as learning rates increased, up to learning rates of approximately 0.55 (Fig. 2e). Crucially, this optimal alpha was above the highest learning rate shown by any of our participants in any recipient condition, meaning a higher learning rate on our task was associated with better performance.

### Statistical analysis.

Analysis of group and recipient differences in the fitted model parameters and behavioural data was run in R[89] v3.6.2 with R Studio[90]. We used a robust linear mixed-effect model (RLMM; rlmer function; robustlmm package[91] to predict learning rates and generalised linear mixed-effects models (glmer function; lme4 package[92]) for the trial-by-trial data (binary outcome of choosing the high vs. low reward option). We used (robust/generalised) linear mixed-effects models as these account for the within-subject nature of the recipient manipulation and do not rely on parametric assumptions. In addition, unlike an ANOVA model or omnibus test, an RLMM generates coefficients (with confidence intervals and significance values) for terms that compare pairs of factor levels (e.g., self vs. other). This approach is more informative than an ANOVA when the factor has more than two levels, which is our case as the recipient factor has three levels (self, other and no one). Each linear mixed-effects model had fixed effects of age group, recipient (self, other, no one) and their interaction, plus a random subject-level intercept. Analysis of trial-by-trial choices also included the trial number in the fixed terms, interacting with recipient and group (including the three-way

interaction), and in the random terms, interacting with the recipient. In the analysis of learning rates controlling for IQ test performance, standardised scores on the WTAR were also included as a fixed term (Supplementary Table 8). Correlations of learning rates with self-reported psychopathic traits (Supplementary Tables 4 and 5) and neuropsychological measures (Supplementary Table 9) were calculated with Spearman's Rho nonparametric tests. Comparisons between independent correlations were calculated with $Z$ tests (paired.r function; psych package[93]) To control for IQ test performance and executive function in the associations between older adults' prosocial learning rates and self-reported psychopathic traits, we ran partial correlations each controlling for one of WTAR, ACE memory and ACE attention scores (Supplementary Table 10).

For simple and post hoc comparisons, we used two-sided nonparametric tests as outcome variables that violated normality assumptions. Effect sizes and confidence intervals for paired and independent nonparametric comparisons were calculated using the cohens_d and wilcox_effsize functions, respectively, from the rstatix package[94]. Bayes factors ($BF_{01}$)[95] for non-significant results were calculated using nonparametric paired and independent $t$ tests in JASP[95] v0.14.1 with the default prior. $BF_{01}$ corresponds to how many times more likely the data are under the null hypothesis of no difference than under the alternative hypothesis that there is a difference. A $BF_{01} > 3$ (equal to $BF_{10} < 1/3$) is considered substantial evidence in favour of the null hypothesis whereas a $BF_{01}$ between 1/3 and 3 indicates the data cannot clearly differentiate between hypotheses[96]. Median learning rates and their standard errors for plotting were calculated using bootstrapping with 1000 samples.

For the analysis of indirect paths, we used the mediate function (mediation package[97]) combined with robust linear models (rlm function, MASS package[98]). This method estimates the unstandardised indirect effects, with 95% confidence intervals and significance, through a bootstrapping procedure with 10,000 bootstrapped samples. The outcome in the models was relative prosocial learning rate ($\alpha_{other} - \alpha_{self}$), the predictor was age group, and the indirect path was through self-reported psychopathic traits on the core affective-interpersonal subscale of the SRP-IV-SF. We calculated two models. The first, a standard mediation model, included an indirect path from age group to relative prosocial learning rate via self-reported psychopathic traits (Fig. 5a). The second included an interaction between age group and self-reported psychopathic traits in predicting relative prosocial learning rate to allow the possibility of a statistically moderated mediation[99]. Specifically, this type of moderated mediation examines whether the indirect path (in this case via self-reported psychopathic traits) on the outcome ($\alpha_{other} - \alpha_{self}$) is moderated by the predictor (age group)[100,101]. In other words, self-reported psychopathic traits could explain some of the variances in relative prosocial learning for one age group but not the other (Fig. 5b).

**Reporting summary**. Further information on research design is available in the Nature Research Reporting Summary linked to this article.

## Data availability

The data generated in this study are available on the Open Science Framework at https://doi.org/10.17605/osf.io/xgw7h. Source data are provided with this paper.

## Code availability

Code for modelling and analysis is available at https://doi.org/10.17605/osf.io/xgw7h.

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

## Acknowledgements

This work was supported by a Medical Research Council Fellowship (MR/P014097/1), a Christ Church Junior Research Fellowship, a Christ Church Research Centre Grant, and a Jacobs Foundation Research Fellowship to P.L.; a Wellcome Trust Principal Fellowship to M.H.; NIHR Biomedical Research Centre, Oxford. The Wellcome Centre for Integrative Neuroimaging is supported by core funding from the Wellcome Trust (203139/Z/16/Z). We thank Ellena Crane for assistance with data collection. We are grateful to Matthew Apps and Miriam Klein-Flugge for helpful disucssions and to Craig Neumann for assistance with the Self-Report Psychopathy Scale. We are also grateful to our colleagues who acted as the other participant during the study.

## Author contributions

P.L. designed the study. A.A., L.H., D.D. and P.L. collected the data. J.C., M.W. and P.L. analysed the data. J.C., M.W., M.H. and P.L. wrote the paper.

## Competing interests

The authors declare no competing interests
