## [Peer Review File · Nature Communications]

Reviewers' comments:

Reviewer #1 (Remarks to the Author):

This is one of those studies that I wish I had thought of! It's such a great idea to evaluate adult age differences in learning for oneself versus others given the decision making findings of slower learning with age and the motivation findings of adults social motivation being intact or enhanced in older age. We have related studies in progress in our own lab (with completely different designs) but nothing is finished yet, so I have no immediate conflicts. I have no major concerns with the study or manuscript. I detail several relatively minor comments below. Most of them have to do with better acknowledging relevant literature.

I don't think it's accurate to say "largely unknown how ageing affects social functioning". Check out the work of the late Fredda Blanchard-Fields on interpersonal problem solving and the Social Cognition and Aging edited volume by Fredda and Tom Hess. This is a large, relevant literature. No need to review it all of course, but acknowledging this area would improve the setup.

In the section on RL and aging, the Mell et al study is detailed which is great but there's been a dozen studies since then. Ben Eppinger and Dorothea Hammerer have some great reviews of RL and aging that could be cited.

There are two studies that inspired our own line of new work on social reward and aging that could be integrated here. One is an fMRI study by Rademacher, Salama, Gründer, & Spreckelmeyer 2013 (<https://academic.oup.com/scan/article/9/6/825/1666365>) showing an age by reward domain (money vs abstract socioemotional reward) interaction in the striatum. The other is a procedural learning study by Gorlick, Giguère, Glass, Nix, Mather, & Maddox, 2013 (<https://psycnet.apa.org/record/2012-30607-001>) showing the benefits of socioemotional feedback for a different kind of learning in older age.

The first study we did in this space might also be relevant. Kendra Seaman, Gorlick, et al showing an interaction between monetary and social rewards in time and probability discounting tasks (<https://psycnet.apa.org/doiLanding?doi=10.1037%2Fpag0000131>).

I'm trying to be less obnoxious about it, but the convenience sampling always bothers me. I don't think it's reasonable to compare a largely university student sample to a community sample of older adults. There are too many confounds. This can be especially problematic in reward tasks when the students are participating to reduce a credit requirement and older adults are making money - independent of the task incentives which may be matched. Thank you for not having that issue here. The matched payments is ideal. It's too late to deal with this sampling issue now obviously but some additional acknowledgement in the discussion might be warranted. It's also more work and less power for a direct age comparison, but too many studies have neglected midlife. A fully community sample that covers adulthood would be ideal for future studies.

Older adults clearly learned more slowly in general which is consistent with most of the published RL studies. It looks like they were catching up with the young potentially after 16 trials. Any data beyond that to see if that's true? Just wondering when/if they reach the same level of performance.

The psychopathy part of the paper could use some different framing/explaining. This is not a

clinical sample and that is made clear. However, some readers might think, ah the young people here are psychopaths and the older aren't. That's not an interesting mechanism, it's a confound of clinical young people and healthy old people. I don't think that's what this is though. Rather, there is a scale that measures psychopathy at the extremes but in normal populations can assess antisociality and interpersonal emotion. I would make this very, very clear and reduce the use of the word psychopathy throughout but especially in the abstract.

Thank you for the inclusion of CIs around every effect. It's refreshing. I tend to prefer standardized effects (especially coming from a lab where we do lots of meta-analysis) but the instances where unstandardized effects are reported really work here. For example, showing the .024 difference in alpha for the self/other effect is very concrete and interpretable. I have no suggestions for changes in the reporting of results.

The figures here are all so beautiful and so clear. Well done.

The discussion mentions prediction error and aging. There are qualifiers of those effects. Both Nichole Lighthall (<https://www.jneurosci.org/content/38/39/8453.short>) and a group from Yael Niv's lab (<https://www.jneurosci.org/content/40/5/1084.abstract>) have data showing intact RPE in aging brains. Also, the first papers on this by me and Ben have pretty small effects on the PE age differences.

Related to neural mechanisms, we've really been trying to figure this out. How do older adults do "better" in tasks that have more goal-relevant features when they perform worse in a more standard version of the task? That's not the exact effect you have here but it's related. The early PET data made us think there should be some shift away from dopamine-mediated reward processing in older age. But our recent digging into the literature and new data suggests there could be dopamine adaptability in older age. For example, Teresa Karrer has a meta-analysis showing intact dopamine synthesis across adulthood (<https://www.sciencedirect.com/science/article/pii/S0197458017301616>) but the receptor declines are large and undeniable so how does that dopamine have any post-synaptic influence? Kendra Seaman did a PET study after that showing some areas of the brain with preserved receptor availability so there could be relatively intact dopamine pathways into very old age (<https://onlinelibrary.wiley.com/doi/abs/10.1002/hbm.24585>). This might not be relevant to the current paper but if you all continue in this area – and I really hope you do – this could be relevant.

Finally, my apologies for taking so long to review this. If you've been waiting, it's my fault. I declined reviews for months but was tempted into this one by the title. It took me too long to actually get to it, so apologies for the delay.

Well done. Can't wait to cite this work.

Gregory Samanez-Larkin

Reviewer #2 (Remarks to the Author):

This study addresses the important and timely question of aging effects in prosocial learning. To this end the authors used a prosocial learning task in 152 participants. The authors conclude that older adults have reduced self-bias, and a relatively increase in the rate at which they learned about actions to help others. Also, psychopathic traits reduced in older

adults, and related negatively to prosocial learning rates. These findings may have implications for the understanding of reinforcement learning mechanisms in healthy ageing. I very much enjoyed reading this well-written paper. They thoroughly and transparently present their modeling work, and include extensive modeling and parameter-based simulations. In sum, the authors present an interesting research paradigm with an innovative modeling approach. I believe their findings provide new insight into the ability versus motivation discussion of age-related change in reinforcement learning. I hope my comments will strengthen this paper further.

My main concern is whether the results of the authors not tell us that the most important finding is the diminishing of self-relevant learning (see also Figure 3a/b). In the main text, the authors reveal planned comparisons between self-other and self-no-one conditions in age-related differences in learning rates. The results they describe (line 249-280) highlight specific age*condition effects when comparing self-other, but not when comparing self-no-one. However, the unpacking of this interaction does not yet show that prosocial learning is persevered specifically in older adults. I believe the authors use the planned within-subject comparisons (ie., pairwise contrasts between conditions per age groups) to make a case for that. However, stating these pairwise contrast without any interactions is somewhat hard to interpret. I do see that the relative change is the largest in the prosocial condition (although relative to what is not always specified, which can be improved), but the absence of interactions -as far as I understand- may be problematic. I hope the authors can convince me otherwise and/or carefully phrase their interpretation.

In addition, I was wondering how the authors current focus with psychopathy fits with their previous focus on empathy. The authors previously established relationships with empathy in their 2016 paper, and it would be interesting to see a) those effects replicated, and b) present a framework why psychopathy would be a more interesting variable from an aging perspective than empathy. Also, I might have missed this, but did the authors also check - and exclude- for outliers in psychopathy scores as they did for other variables such as learning rates?

The discussion could be further supported by highlighting the advantages (and challenges) of the authors' model-based approach. For instance, one thing that stands out here is that their model-based parameters are more sensitive to effects of aging on self and prosocial learning than general performance measures. In the supplementary results I would consider it informative to see the beta-distributions per age group

Finally, the authors mention the limits of testing different age groups, versus studying ageing continuously. Space permitting, I would love to see an explorative analysis on continuous age-related change within age groups, if age distribution, as well as the sample sizes, allow such analyses

Reviewer #3 (Remarks to the Author):

This manuscript examines prosocial reinforcement learning across aging. The central finding is that prosocial learning rates are similar in older and younger adults despite a decline in learning rates for oneself in older adults. This is interpreted in light of changes in prosocial motivation over the lifespan.

The methods and analyses in this paper are appropriate, quite nicely performed, and openly described. The question of changes in learning over the lifespan is generally an interesting one. That said, I have some concerns about the inferences drawn in this paper.

1) One concern is that higher learning rates are interpreted as reflecting stronger prosocial motivation. Although this is possible, I am not convinced that this is sufficiently justified; higher learning rates are not always better (Zhang et al., 2020, SCAN). In particular, the fact that learning rates were positively related to psychopathy in younger adults suggests that other processes besides prosocial concern were likely at play.

Relatedly, the discussion section suggests: "Our findings suggest that despite declines in learning ability associated with ageing, motivation could play a role in preserving learning to help others."

However, this is not directly tested. The authors note that prosocial and self-oriented learning involve distinct neural mechanisms; as a result, another possibility is that mechanisms related to self-oriented learning decline whereas mechanisms related to other-oriented learning do not. This interpretation would not relate to prosocial motivation. Of course, this finding might still have interesting implications given the importance of social connections in older adults, but would require a distinct interpretation.

2. A second concern is that two central claims are written in ways that can easily be misunderstood. The abstract states: "Strikingly however, older adults showed reduced self-bias, with a relative increase in the rate at which they learnt about actions that helped others, compared to themselves. Moreover, we find evidence that these group differences are associated with changes in psychopathic traits over the lifespan."

Although "a relative increase in the rate at which they learnt about actions that helped others" is technically true, this makes it sound like prosocial learning increased, whereas the primary finding is that self-oriented learning rates decreased.

Similarly, "these group differences are associated with changes in psychopathic traits over the lifespan" sounds very much like it is describing a mediation model where psychopathic traits change, and this change is associated with group differences in learning rates. However, this is not the pattern of results described in the paper; group differences in learning rates were not associated with group differences in psychopathic traits. Instead, psychopathic traits had different patterns of correlation with learning rates within groups.

3) The primary analyses examine the interaction of age x target across self and other. However, it's not obvious to me why the analysis should be restricted to these two targets, instead of examining the trend across all three conditions. That is, is there less of a reduction in learning rate for "other" than for "no one" among older adults? This interaction would be important to inform whether results are truly due to changes in prosocial learning as opposed to changes in self-learning mechanisms alone.

Minor points:

1. The introduction states: "It is largely unknown how ageing affects social functioning, despite the critical importance of this question." This is too broad a claim and should more specifically address prosocial learning in aging. The authors might also elaborate on how prosocial learning contributes to our understanding of social functioning in aging.
2. When describing overall group differences in psychopathy, it would help if the authors can

report the means directly in the text.

3. Fig 3a appears to show far less variability in other-learning rates relative to self or no one. How do the authors interpret this?

Reviewers' comments:

Reviewer #1 (Remarks to the Author):

This is one of those studies that I wish I had thought of! It's such a great idea to evaluate adult age differences in learning for oneself versus others given the decision making findings of slower learning with age and the motivation findings of adults social motivation being intact or enhanced in older age. We have related studies in progress in our own lab (with completely different designs) but nothing is finished yet, so I have no immediate conflicts. I have no major concerns with the study or manuscript. I detail several relatively minor comments below. Most of them have to do with better acknowledging relevant literature.

Response: Thank you so much for your positive feedback and enthusiasm about our work. We really appreciate your time to review our work and your valuable comments that have helped us to improve our manuscript.

R1.1: I don't think it's accurate to say "largely unknown how ageing affects social functioning". Check out the work of the late Fredda Blanchard-Fields on interpersonal problem solving and the Social Cognition and Aging edited volume by Fredda and Tom Hess. This is a large, relevant literature. No need to review it all of course, but acknowledging this area would improve the setup.

Response: We apologise for our lack of clarity here and we have amended our wording. We have put references in APA format throughout this document to show details of citations (page 3):

“Senescence is associated with a multitude of changes including declines in cognitive functioning and perception, but perhaps preservation of affective processing and social cognitive abilities (Blanchard-Fields, 2007; Blanchard-Fields & Hess, 1999; Samanez-Larkin & Knutson, 2015). However, less is known about how ageing affects social behaviour, despite the critical importance of this question. Social isolation has been found to be as damaging to physical health as smoking or excessive drinking (Holt-Lunstad et al., 2015). Social behaviours that benefit others – prosocial behaviours – are vital for maintaining social bonds and relationships (Fehr & Fischbacher, 2003) across the lifespan. In addition to the benefits for others, prosociality has been linked to improved life satisfaction (Buchanan & Bardi, 2010), mental wellbeing (Raposa et al., 2016), and physical health (Post, 2005) for the person being prosocial, all of which could contribute to healthy ageing. A key aspect of prosocial behaviour is the ability to learn associations between our own actions and outcomes for other people (Lockwood et al., 2016).”

R1.2: In the section on RL and aging, the Mell et al study is detailed which is great but there's been a dozen studies since then. Ben Eppinger and Dorothea Hammerer have some great reviews of RL and aging that could be cited.

Response: Thank you for highlighting this work. We have now added these additional references to our introduction (page 4):

“Although essential for successful adaptive behaviour, several studies suggest that our propensity for reinforcement learning declines in later life (Samanez-Larkin & Knutson, 2015). Compared to younger adults, older adults show learning impairments particularly when action-outcome associations are probabilistic

(Eppinger et al., 2011) or reverse (Mell et al., 2005). Age-related declines in learning ability have been linked to functional and structural changes in frontostriatal circuits (Samanez-Larkin et al., 2012, 2014) and dopamine transmission (Eppinger et al., 2011; Hammerer & Eppinger, 2012), which shows a significant age-related decrease (Bäckman et al., 2006; Dreher et al., 2008; Li et al., 2010) and has a key role in coding prediction errors (Glimcher, 2011; Schultz, 2013, 2016)."

R1.3: There are two studies that inspired our own line of new work on social reward and aging that could be integrated here. One is an fMRI study by Rademacher, Salama, Gründer, & Spreckelmeyer 2013 (<https://academic.oup.com/scan/article/9/6/825/1666365>) showing an age by reward domain (money vs abstract socioemotional reward) interaction in the striatum. The other is a procedural learning study by Gorlick, Giguère, Glass, Nix, Mather, & Maddox, 2013 (<https://psycnet.apa.org/record/2012-30607-001>) showing the benefits of socio-emotional feedback for a different kind of learning in older age.

The first study we did in this space might also be relevant. Kendra Seaman, Gorlick, et al showing an interaction between monetary and social rewards in time and probability discounting tasks (<https://psycnet.apa.org/doiLanding?doi=10.1037%2Fpag0000131>).

Response: Thank you for bringing our attention to this interesting work. We have incorporated it into our discussion where we consider our findings in the broader context of social cognition and motivation (page 22):

"The idea that social motivations become more influential in learning and decision-making with age has also been suggested based on studies of social rewards such as smiling faces or hypothetical time spent with social partners (Gorlick et al., 2013; Rademacher et al., 2014; Seaman et al., 2016). Taken together, this work suggests that strategies to support healthy ageing might benefit from leveraging potentially preserved social motivations."

R1.4: I'm trying to be less obnoxious about it, but the convenience sampling always bothers me. I don't think it's reasonable to compare a largely university student sample to a community sample of older adults. There are too many confounds. This can be especially problematic in reward tasks when the students are participating to reduce a credit requirement and older adults are making money - independent of the task incentives which may be matched. Thank you for not having that issue here. The matched payments is ideal. It's too late to deal with this sampling issue now obviously but some additional acknowledgement in the discussion might be warranted. It's also more work and less power for a direct age comparison, but too many studies have neglected midlife. A fully community sample that covers adulthood would be ideal for future studies.

Response: Thank you for your positive feedback on our approach to matched payments. We completely agree that matching of payments is a particularly important feature in incentivized tasks comparing groups. Regarding sampling, our young group incorporated both students and members of the community, and we were careful to match the groups as closely as possible in terms of gender and education. Many of the older adults also came from university databases so they were perhaps more matched than studies using young samples purely from university populations and older adults not from university populations. To highlight our sampling strategy more clearly in the manuscript we have now provided further information in the methods section. We have also added a note in the discussion for future studies to compare a full community sample, as well as including the whole adult lifespan:

Methods (page 25) -

“We recruited 80 young participants and 80 older participants using the same recruitment methods in order to match the samples as closely as possible. Participants were recruited from university databases, which included students and members of the community, social media, and adverts in local newspapers. We excluded anyone who was currently studying or had previously studied psychology and no one took part for course credit. Additional exclusion criteria were previous or current neurological or psychiatric disorder, non-normal or non-corrected to normal vision and, for the older sample, scores on the Addenbrooke’s Cognitive Examination that indicate potential dementia (cut-off score 82) (Hsieh et al., 2013).”

Discussion (page 24) -

“While our procedure and task have many benefits, it is important to also recognise limitations. To test for age-related differences in prosocial learning, we recruited a group of older adults and a group of younger adults. This increases power to detect differences, but we are unable to assess at what age or how quickly changes take place. Also, while our age groups were matched on years of education and IQ, the recruitment from university databases or issues around self-selection may mean that the levels of education and IQ in our sample are not completely representative of the general population. Further studies could include samples recruited entirely from the community and participants across the whole adult lifespan.”

R1.5: Older adults clearly learned more slowly in general which is consistent with most of the published RL studies. It looks like they were catching up with the young potentially after 16 trials. Any data beyond that to see if that's true? Just wondering when/if they reach the same level of performance.

Response: Thank you for this suggestion. In order to sample enough conditions within the testing session each stimulus was presented 16 times per block. We did not include more stimulus repetitions to prevent the task getting too long. We agree it is a really interesting point as to whether older adults might eventually catch up with the performance of younger adults (and if so, if this might be driven by motivational aspects). We have added these ideas in the discussion (page 24):

“Future studies should also assess the timescale after which older participants reach similar ceiling levels of performance as younger adults, or whether they are never able to reach the same level of performance. Of similar importance is the question of whether older adults may be able to sustain higher levels of motivation over an extended period of time compared to younger adults, possibly compensating for slower learning speeds. In this study, due to time constraints and the presence of several conditions, we are only able to derive conclusions from a limited time window.”

R1.6: The psychopathy part of the paper could use some different framing/explaining. This is not a clinical sample and that is made clear. However, some readers might think, ah the young people here are psychopaths and the older aren't. That's not an interesting mechanism, it's a confound of clinical young people and healthy old people. I don't think that's what this is though. Rather, there is a scale that measures psychopathy at the extremes but in normal populations can assess antisociality and interpersonal emotion. I

would make this very, very clear and reduce the use of the word psychopathy throughout but especially in the abstract.

Response: We appreciate you raising this point. We have now revised the manuscript to ensure that all mentions of this measure in normal populations, particularly our sample, say 'psychopathic traits' and not 'psychopathy'. We have also added extra information in both the introduction and results section to explain further details about the Self-Report Psychopathy Scale we used:

Introduction (page 5) -

“At the extreme, psychopathy is a severe personality condition linked to poor life outcomes, violence, and criminality (Asscher et al., 2011; Blais et al., 2014; Leistico et al., 2008). However, several studies suggest similar behavioural and neural profiles between community samples with high levels of psychopathic traits (Seara-Cardoso et al., 2012) and those with clinical diagnoses of psychopathy, consistent with the Research Domains of Criteria (RDoC) conceptualisation of a dimensional approach to psychiatry (Insel, 2014). This RDoC approach suggests that psychiatric disorders can be thought of as dimensional, rather than categorical, constructs. In a similar vein, psychopathic traits can be captured on a continuum spanning clinical samples and the general population, with a range of scores on that continuum for healthy people. Self-report measures of psychopathic traits that mirror the latent structure of clinical psychopathy measures, comprising antisocial and interpersonal dimensions, are available to use in samples from the general population (Paulhus et al., 2017).”

Results (page 15) -

“Therefore, we also asked participants to complete the Self-Report Psychopathy Scale (SRP-IV-SF) (Paulhus et al., 2017). The SRP is a measure of psychopathic traits in healthy samples that assesses traits linked to clinical psychopathy, such as antisociality and interpersonal affect. The measure robustly captures the latent structure of clinical psychopathy to enable parallels can be drawn between normal individual differences in the community and clinical samples (see Methods).”

R1.7: Thank you for the inclusion of CIs around every effect. It refreshing. I tend to prefer standardized effects (especially coming from a lab where we do lots of meta-analysis) but the instances where unstandardized effects are reported really work here. For example, showing the .024 difference in alpha for the self/other effect is very concrete and interpretable. I have no suggestions for changes in the reporting of results.

The figures here are all so beautiful and so clear. Well done.

Response: Thank you for such positive feedback on the reporting of our results and our figures.

R1.8: The discussion mentions prediction error and aging. There are qualifiers of those effects. Both Nichole Lighthall (<https://www.jneurosci.org/content/38/39/8453.short>) and a group from Yael Niv's lab (<https://www.jneurosci.org/content/40/5/1084.abstract>) have data showing intact RPE in aging brains. Also, the first papers on this by me and Ben have pretty small effects on the PE age differences.

Response: We really appreciate you sending the links to these relevant papers and have added reference to these in our discussion (page 21):

“Research combining models of learning with neuroimaging and pharmacological manipulations suggests ageing reduces the ability to generate reward prediction errors (Eppinger et al., 2011) (but see (Daniel et al., 2020; Lighthall et al., 2018)) due to declines in dopamine functioning (Chowdhury et al., 2013; Nieuwenhuis et al., 2002) (also see (Karrer et al., 2017; Seaman et al., 2019)).”

R1.9: Related to neural mechanisms, we've really been trying to figure this out. How do older adults do "better" in tasks that have more goal-relevant features when they perform worse in a more standard version of the task? That's not the exact effect you have here but it's related. The early PET data made us think there should be some shift away from dopamine-mediated reward processing in older age. But our recent digging into the literature and new data suggests there could be dopamine adaptability in older age. For example, Teresa Karrer has a meta-analysis showing intact dopamine synthesis across adulthood (<https://www.sciencedirect.com/science/article/pii/S0197458017301616>) but the receptor declines are large and undeniable so how does that dopamine have any post-synaptic influence? Kendra Seaman did a PET study after that showing some areas of the brain with preserved receptor availability so there could be relatively intact dopamine pathways into very old age (<https://onlinelibrary.wiley.com/doi/abs/10.1002/hbm.24585>). This might not be relevant to the current paper but if you all continue in this area – and I really hope you do – this could be relevant.

Response: Thank you for highlighting all this relevant literature and for the useful discussion. We find the idea that older adults do better in tasks with more goal directed features intriguing and particularly how it might fit with research suggesting a distinction between motivation and ability. We were a little hesitant to go too much into the possible neural correlates given the present study is behavioural, despite probing mechanisms that have a clear biological link within dopamine signalling more broadly. We would love to follow up on these ideas in future work. We have added these two references at the point in the discussion we mention age-related declines in dopamine function (page 21):

“Research combining models of learning with neuroimaging and pharmacological manipulations suggests ageing reduces the ability to generate reward prediction errors (Eppinger et al., 2011) (but see (Daniel et al., 2020; Lighthall et al., 2018)) due to declines in dopamine functioning (Chowdhury et al., 2013; Nieuwenhuis et al., 2002) (also see (Karrer et al., 2017; Seaman et al., 2019)).”

Finally, my apologies for taking so long to review this. If you've been waiting, it's my fault. I declined reviews for months but was tempted into this one by the title. It took me too long to actually get to it, so apologies for the delay.

Response: Thanks so much again for your time and helpful comments that have improved our manuscript!

Well done. Can't wait to cite this work.

Gregory Samanez-Larkin

Reviewer #2 (Remarks to the Author):

This study addresses the important and timely question of aging effects in prosocial learning. To this end the authors used a prosocial learning task in 152 participants. The authors conclude that older adults have reduced self-bias, and a relatively increase in the rate at which they learned about actions to help others. Also, psychopathic traits reduced in older adults, and related negatively to prosocial learning rates. These findings may have implications for the understanding of reinforcement learning mechanisms in healthy ageing. I very much enjoyed reading this well-written paper. They thoroughly and transparently present their modeling work, and include extensive modeling and parameter-based simulations. In sum, the authors present an interesting research paradigm with an innovative modeling approach. I believe their findings provide new insight into the ability versus motivation discussion of age-related change in reinforcement learning. I hope my comments will strengthen this paper further.

Response: We really appreciate your positive comments about our work, as well as your time providing suggestions for improving the paper, thank you.

R2.1: My main concern is whether the results of the authors not tell us that the most important finding is the diminishing of self-relevant learning (see also Figure 3a/b). In the main text, the authors reveal planned comparisons between self-other and self-no-one conditions in age-related differences in learning rates. The results they describe (line 249-280) highlight specific age*condition effects when comparing self-other, but not when comparing self-no-one. However, the unpacking of this interaction does not yet show that prosocial learning is persevered specifically in older adults. I believe the authors use the planned within-subject comparisons (ie., pairwise contrasts between conditions per age groups) to make a case for that. However, stating these pairwise contrast without any interactions is somewhat hard to interpret. I do see that the relative change is the largest in the prosocial condition (although relative to what is not always specified, which can be improved), but the absence of interactions -as far as I understand- may be problematic. I hope the authors can convince me otherwise and/or carefully phrase their interpretation.

Response: We apologise for any lack of clarity in the results reporting and have made edits to more clearly explain our analysis and the rationale behind it.

To assess interactions between age group and recipient we used a robust linear mixed-effects model (RLMM), since this approach does not rely on parametric assumptions and accounts for the within-subject nature of the recipient manipulation. Additionally, unlike an anova model, this model generates coefficients (with confidence intervals and significance values) for terms that compare pairs of factor levels (for example self vs. other) (Bates et al., 2015). This approach is more informative than an anova when the factor has more than two levels, which is our case as the recipient factor has three levels (self, other, no one). For example, the anova result described below shows there was an interaction between age group and recipient generally, but not which recipient conditions drive this interaction. In contrast, our RLMM generates results for the age group * recipient [self vs. other] interaction and the age group * recipient [self vs. no one] interaction. In all analyses using any form of linear model and a predictor with more than 2 levels, the number of these terms (for both main effects and interactions) is always the number of levels – 1 (Fox, 2015), so 2 in our case. These are self vs. other and self vs. no one because the self condition is the reference group. We have now included further information on the rationale for using a RLMM in the methods section.

As the format of the results generated by our RLMM is slightly different to an anova model, as described above, we additionally ran the same model through a method that treats it as a robust two-way anova (mixed between-within subject design) using a bootstrapping method to test the overall group*recipient interaction (sspb function from the WRS2 package (Mair & Wilcox, 2020), 10,000 iterations). This shows a significant overall interaction ($p = 0.045$). We have not currently added this to the paper as it represents the same analysis as our RLMM and cannot be interpreted without the age * pairs of recipient conditions interactions and the post hoc comparisons we report. However, we would be happy to include it if you think it would be useful for readers.

Crucially, we also now report the RLMM interaction term for the non-significant age group * recipient [self vs. no one] interaction. Combined with the significant age group * recipient [self vs. other] interaction, this shows prosocial learning, relative to learning for the self, differs between the age groups in a way that learning for no one, relative to learning for the self, does not. Although these interactions (from the RLMM including all three levels of the recipient condition) are specific to pairs of recipient conditions, post hoc pairwise contrasts are needed to fully understand how the pattern of learning rates across recipients differs between young and older participants. We have edited the wording in the results section to better describe the fact that we consider the within-subject and between-subject comparisons together in generating our conclusions.

Following on from the significant interaction of recipient [self vs other] and age, we describe prosocial learning as ‘preserved’ in older adults because our between-group comparison showed strong Bayesian evidence in support of no difference between the age groups in prosocial learning rates. In contrast, learning rates for the self were significantly lower in older, compared to younger adults. A decrease in learning rates during self-relevant learning with age has been shown in multiple previous studies, as outlined in our introduction. Against this background of robust evidence that self-relevant learning is disrupted with age, we believe the preservation of prosocial learning rates in older adults at the same level as young adults is striking. The comparison of learning rates for no one between the age groups did not show a significant difference or evidence for no difference, so we do not interpret this as the same or different between young and older adults. Results from comparing other vs. no one in each group show that older adults had higher learning rates for other than no one whereas young adults did not differentiate between other and no one. These results based on the no one condition are important as they show older adults still distinguished between other and no one and therefore the age-related decline in self-bias were not simply due to older adults no longer being able to process who receives the reward.

We have rewritten the main part of our results section to make it clearer for each pairwise comparison what the difference (or lack of difference) is relative to and to make it clear that our RLMM included all three recipient conditions (self, other, no one) (page 13-14):

*“We analysed the condition-specific learning rates from our best fitting computational model using a robust linear mixed-effects model (RLMM; see Methods). The RLMM fixed effects were age group (young, older), recipient (self, other, no one), as well as the age group * recipient interaction. While this RLMM includes all three recipient conditions, it generates coefficients (main effect and the interaction with age group) contrasting pairs of recipient conditions – [self vs. other] and [self vs. no one]. These are more interpretable than an omnibus test that would not show which recipient conditions were driving an effect or interaction. We followed up these results with*

planned comparisons, between the older and younger group in each recipient condition, and between pairs of recipient conditions within each age group.

Across age groups, participants showed a higher learning rate when rewards were for themselves, compared to for another person (recipient [self vs. other]: $b=-0.024$ [-0.034, -0.014], $z=-4.79$, $p<0.001$). Importantly however, this pattern differed between age groups. The difference between learning rates for self and other was reduced in older compared to younger adults (recipient [self vs. other] * age group interaction: $b=0.016$ [0.002, 0.030], $z=2.29$, $p=0.02$). Between-group comparisons showed older adults learnt more slowly for themselves compared to younger adults ($W=3512$, $Z=-2.63$, $r_{(150)}=0.22$ [0.06, 0.36], $p=0.009$). However, prosocial learning was preserved, with a Bayes factor suggesting strong evidence of no difference in α_{other} between young and older adults ($W=3042$, $Z=-0.86$, $r_{(150)}=0.07$ [0.00, 0.24], $p=0.39$, $BF_{01}=4.26$). Within-subject comparisons of α_{self} and α_{other} in each age group showed that young adults had higher learning rates for themselves, relative to another person ($V=659$, $Z=-4.04$, $r_{(75)}=0.47$ [0.26, 0.63], $p<0.001$). In contrast, older adults showed no significant difference between learning rates for self and other ($V=1150$, $Z=-1.45$, $r_{(75)}=0.17$ [0.01, 0.38], $p=0.15$, $BF_{01}=1.08$).

As expected, across age groups learning was slower for no one than self (recipient [self vs. no one]: $b=-0.023$ [-0.033, -0.013], $z=-4.57$, $p<0.001$). Unlike α_{self} vs. α_{other} , learning for self compared to no one did not interact with age group (recipient [self vs. no one] * age group interaction: $b=0.008$ [-0.006, 0.022], $z=1.15$, $p=0.25$). Within-subject comparisons between α_{self} and $\alpha_{no\ one}$ in each age group showed that both groups learnt preferentially for themselves compared to no one (young adults: $V=928$, $Z=-2.62$, $r_{(75)}=0.30$ [0.07, 0.51], $p=0.009$; older adults $V=901$, $Z=-2.76$, $r_{(75)}=0.32$ [0.09, 0.53], $p=0.006$). There was no significant difference between the age groups in $\alpha_{no\ one}$ but also no evidence in support of the null ($W=3241$, $Z=-1.61$, $r_{(150)}=0.13$ [0.01, 0.29], $p=0.11$, $BF_{01}=2.04$).

Considering differences in learning between α_{other} and $\alpha_{no\ one}$, young adults did not differentiate between another person and no one, with strong Bayesian evidence for no difference ($V=1533$, $Z=-0.57$, $r_{(75)}=0.07$ [0.00, 0.31], $p=0.57$, $BF_{01}=5.08$). In contrast, older adults had higher learning rates for another person, compared to no one ($V=976$, $Z=-2.37$, $r_{(75)}=0.27$ [0.05, 0.49], $p=0.02$). Crucially, this shows that older adults' lack of differentiation between self and other was not simply because they were insensitive to the recipient condition.

Finally, we also observed an effect of age on both learning rates overall and temperature parameters. Older adults showed slower learning overall compared to younger adults ($b=-0.019$ [-0.028, -0.009], $Z=-3.73$, $p<0.001$) and higher levels of exploration of choice options (median β young: 0.05, older: 0.19, $W=1511$, $Z=-4.89$, $r_{(150)}=0.40$ [0.26, 0.53], $p<0.001$; Supplementary Figure 1).

In summary, older adults prosocial learning was preserved at the same rate as young adults, despite age-related declines in self-relevant learning rates. In other words, young adults showed a self-bias in learning, but older adults distinguished between themselves and others significantly less than the young participants. Only older adults, not young adults, distinguished between rewards for another person and no one.”

Methods (note we have put references in APA format throughout this document to show details of citations, page 32-33) -

“We used a robust linear mixed-effect model (RLMM; rlmmer function; robustlmm package (Koller, 2016) to predict learning rates and generalised linear mixed-effects models (glmer function; lme4 package (Bates et al., 2015)) for the trial-by-trial data (binary outcome of choosing the high vs. low reward option). We used (robust / generalised) linear mixed-effects models as these account for the within-subject nature of the recipient manipulation and do not rely on parametric assumptions. Additionally, unlike an anova model or omnibus test, an RLMM generates coefficients (with confidence intervals and significance values) for terms that compare pairs of factor levels (for example self vs. other). This approach is more informative than an anova when the factor has more than two levels, which is our case as the recipient factor has three levels (self, other, no one).”

R2.2a: In addition, I was wondering how the authors current focus with psychopathy fits with their previous focus on empathy. The authors previously established relationships with empathy in their 2016 paper, and it would be interesting to see a) those effects replicated, and b) present a framework why psychopathy would be a more interesting variable from an aging perspective than empathy.

Response: Thank you for this suggestion. We agree that empathy could also be an interesting variable in this context, but to avoid issues of multiple comparisons and because the current work stems from a completely separately funded research project investigating ageing and individual differences in psychopathology, we focus on psychopathic traits in this study. Whilst the senior author of this paper and the first author of the 2016 paper are the same, the work we present here was with a distinct team of researchers and where the focus was not to test modulation by trait empathy.

In addition, we do believe that the association between psychopathic traits and prosocial behaviours are a more interesting variable to investigate from an ageing perspective. Whilst some previous work has focused on empathy and prosocial behaviour in ageing, no existing studies, to our knowledge, have examined associations between prosocial behaviours and psychopathic traits. The measure of psychopathic traits we used maps on closely to the clinical construct of psychopathy in terms of its structure but can be used to assess these traits in community samples. This means that there is a possibility for findings to be more directly translated between those found in clinical samples and in typical individual differences in the healthy population, consistent with the Research Domains of Criteria (RDoC) conceptualisation for understanding psychiatric disorders. This gives the measure a robust psychometric and theoretical foundation to examine how subclinical psychopathology might evolve in ageing. We have added the following to the introduction to further emphasise why measures of psychopathic traits are important for assessing changes in behaviour in ageing (page 5):

“Intriguingly, preliminary evidence suggests that ageing may also be associated with changes in psychopathic traits, which could have important implications for our understanding of an ageing population. Epidemiological studies show that criminal activity increases during adolescence then declines in older adulthood (Lieberman, 2008). Antisocial and aggressive behaviours also significantly decrease in older age, with young adults (age 16-24 years) having the highest rates of homicide (Homicide in England and Wales - Office for National Statistics). Even within violent male offenders, psychopathic traits linked to an antisocial lifestyle are negatively correlated with age (Huchzermeier et al., 2008). In community samples, ageing is associated with a decrease in both the antisocial-lifestyle (antisocial and impulsive behaviours) and affective-interpersonal (lack of empathy and guilt) elements of psychopathic traits (Gill & Crino, 2012). These studies highlight the importance of assessing how differences in psychopathic traits could map on to differences in prosocial behaviours. However, no existing work has examined this question.”

We have also added a note in the discussion that future studies could also link the changes observed here to changes in empathy and acknowledging the previous work with this paradigm (page 24):

“Moreover, previous research has suggested that individual differences in empathy – the ability to vicariously experience and understand others’ affect – might relate to differences in prosocial learning (Lockwood et al., 2016). Empathy is positively associated with affective-interpersonal psychopathic traits and might also relate to motivation to help others (Contreras-Huerta et al., 2020; Lockwood et al., 2017). Further studies could also assess how empathy predicts changes in prosocial learning across the lifespan.”

R2.2b: Also, I might have missed this, but did the authors also check -and exclude- for outliers in psychopathy scores as they did for other variables such as learning rates?

Response: Thank you for raising this. We had not checked for or excluded participants based on scores on the psychopathic traits measure. This is because it is not clear what an ‘outlier’ score on the psychopathic traits might mean when the measure is used to detect individual variation between people across the full range of possible scores. We believe this is different from performance measures such as the learning rate where an outlier learning rate suggests a participant was not completing the task properly. Furthermore, all analyses we ran on the psychopathic traits scores, including correlations with learning rates, were nonparametric tests to account for possible extreme scores.

However, to ensure our results are robust, even to extreme scores on trait measures, we re-ran all of the analyses that include the psychopathic trait scores excluding participants with score more than 3 standard deviations from the mean. Six [3 young, 3 older] participants on the affective interpersonal and 5 [2 young, 3 older] participants on the lifestyle antisocial subscale met this threshold. All the results were the same when removing these participants. As this analysis is post hoc and for the reasons we describe above, we kept the analysis including all participants in the main manuscript, but this additional analysis to the supplement and included reference to this supplementary section in the main text:

Figure 4 legend (page 18) -

“(a) For older adults, levels of psychopathic traits are negatively correlated with prosocial learning rates ($r_s = -0.33$ [-0.52, -0.11], $p = 0.005$, false discovery rate (FDR)

corrected $p=0.03$). **(b)** There is no significant relationship for young adults ($r_s=0.21$ [-0.02, 0.42], $p=0.07$, FDR-corrected $p=0.22$) and the correlation is significantly more negative ($Z=3.28$, $p=0.001$) for older than young adults. This pattern of results is the same when considering correlations between psychopathic traits and the lack of self-bias in learning ($\alpha_{other} - \alpha_{self}$; not shown). This measure of prosocial learning is also negatively correlated with psychopathic traits in older adults ($r_s=-0.25$ [-0.45, -0.02], $p=0.03$) but not younger adults ($r_s=0.11$ [-0.12, 0.33], $p=0.36$; difference $Z=2.15$, $p=0.03$). Age-group differences in psychopathic traits and the correlation between α_{other} and psychopathic traits, for older adults only, also remained significant when excluding extreme scores (>3 SDs from the mean) on the psychopathic traits measure (Supplementary Tables 6 & 7). Shaded areas represent 95% confidence intervals.”

Supplement (page 6) -

“Supplementary Table 6. Comparisons of psychopathic traits between groups excluding extreme scores

Subscale	Young mean	Older mean	W	Z	$r_{(df)}$ [95% CI]	p
Affective-interpersonal	23.65	20.19	3411	-3.28	$r_{(142)}=0.27$ [0.12, 0.42]	0.001**
Lifestyle-antisocial	22.55	19.65	3395	-3.04	$r_{(143)}=0.25$ [0.10, 0.40]	0.002**

Note. Extreme scores on the psychopathic traits measure (Self-Report Psychopathy Scale) were defined as those more than 3 standard deviations from the mean, the exclusion criteria applied to learning rates for all analyses. Comparisons are between-group Wilcoxon t-tests. Asterisks represent significance (** $p<0.01$).

Supplementary Table 7. Correlations between learning rates and affective-interpersonal psychopathic traits excluding extreme scores

	Young			Older		
	α_{self}	α_{other}	$\alpha_{no\ one}$	α_{self}	α_{other}	$\alpha_{no\ one}$
r_s	-0.03	0.15	-0.07	0.12	-0.31	0.00
p	0.79	0.20	0.58	0.30	0.008*	0.97
FDR p	0.95	0.60	0.87	0.60	0.05*	0.97

Note. Extreme scores on the psychopathic traits measure (Self-Report Psychopathy Scale affective-interpersonal subscale) were defined as those more than 3 standard deviations from the mean, the exclusion criteria applied to learning rates for all analyses. FDR: false discovery rate correction. Asterisks represent significance (* $p<0.05$).

R2.3a: The discussion could be further supported by highlighting the advantages (and challenges) of the authors' model-based approach. For instance, one thing that stands out here is that their model-based parameters are more sensitive to effects of aging on self and prosocial learning than general performance measures.

Response: Thank you for this suggestion. We have added the following section to the discussion (page 23):

“Our results also support the advantages of a model-based approach for understanding both prosocial behaviour and ageing. The model-based parameters were more sensitive to the effects of interest than general measures of performance on the task. The model comparison process is able to provide important information about how the learning process takes place, which cannot be revealed from performance measures alone. We showed that learning was best represented by a separate learning rate for each recipient. Moreover, this approach is able to capture additional latent parameters that drive behaviour, such as the inverse temperature, which indexes how closely participants follow the stimulus value. We demonstrate that a single inverse temperature parameter best explained behaviour during learning across recipient conditions, despite learning rates being distinct.”

R2.3b: In the supplementary results I would consider it informative to see the beta-distributions per age group

Response: Thank you for this suggestion, we have added a figure showing the beta averages and distributions in each age group to the supplement (page 5):

“Supplementary Figure 1. Comparison of inverse temperature parameters (β) between age groups. Older adults had significantly higher β parameters than young adults, suggesting age is associated with less consistency in choices during the task. Bars show group median, error bars are standard error of the median, the asterisk represents a significant difference from a between-group Wilcoxon t-test ($p < 0.001$); $n = 152$ (75 young, 77 older).”

R2.4: Finally, the authors mention the limits of testing different age groups, versus studying ageing continuously. Space permitting, I would love to see an explorative analysis on continuous age-related change within age groups, if age distribution, as well as the sample sizes, allow such analyses

Response: We agree that it would be really interesting to consider age as a continuous predictor. We looked into running all of the analyses you suggested but unfortunately simply do not have the age distribution or power for these analyses that treat age as a continuous variable within age groups (or across the full sample of course due to the lack of participants

in middle adulthood). We have now expanded on our note on samples for future studies, including recruiting from the full lifespan range, which would allow age to be used as a continuous predictor (page 24):

“While our procedure and task have many benefits, it is important to also recognise limitations. To test for age-related differences in prosocial learning, we recruited a group of older adults and a group of younger adults. This increases power to detect differences, but we are unable to assess at what age or how quickly changes take place. Also, while our age groups were matched on years of education and IQ, the recruitment from university databases or issues around self-selection may mean that the levels of education and IQ in our sample are not completely representative of the general population. Further studies could include samples recruited entirely from the community and participants across the whole adult lifespan.”

Reviewer #3 (Remarks to the Author):

This manuscript examines prosocial reinforcement learning across aging. The central finding is that prosocial learning rates are similar in older and younger adults despite a decline in learning rates for oneself in older adults. This is interpreted in light of changes in prosocial motivation over the lifespan.

The methods and analyses in this paper are appropriate, quite nicely performed, and openly described. The question of changes in learning over the lifespan is generally an interesting one. That said, I have some concerns about the inferences drawn in this paper.

Response: Thank you for taking the time to review our paper and for your positive feedback on our methods, analysis and topic. We appreciate your comments, which have helped us to improve our paper.

R3.1a: 1) One concern is that higher learning rates are interpreted as reflecting stronger prosocial motivation. Although this is possible, I am not convinced that this is sufficiently justified; higher learning rates are not always better (Zhang et al., 2020, SCAN).

Response: Thank you for highlighting this important point. We agree that higher learning rates are not better under all circumstances. Crucially, in our task having a higher learning rate is indeed associated with better performance. We have now conducted two additional control analyses to robustly support the link between learning rates and performance. First, we conducted a simulation experiment determining the optimal learning rate (defined in relation to maximal performance) for our task. In the absence of other local maxima, we found that maximal performance is reached by completing the task with a learning rate of ~0.55. Such a learning rate is higher than in every single participant we tested. Therefore, participants with a learning rate closer to this optimum should perform better in the task. To show that this is also true in our participants, we additionally performed a correlation analyses based on our empirical data and indeed show that participants with a higher learning rate perform better in the task. Overall, while we agree that higher learning rates are not always better (we now highlight this in the paper), we clearly demonstrate that higher learning rates are better in the context of our task. We have edited the following sections.

Methods (note we have put references in APA format throughout this document to show details of citations, page 32) -

“Finally, we conducted a principled simulation experiment to identify the optimal learning rate in our task and examine the link between learning rates and performance. We simulated data from 10,000 participants with learning rates (α) and softmax temperature parameters (β) drawn from beta and gamma distributions respectively, as described above. We bounded the β parameters at 0 and 0.3 to reflect the range shown by our participants. In line with our winning model, simulated participants had a separate learning rate for each recipient condition, and these spanned the full range of possible α values from 0 to 1. This generated 30,000 learning rate values (10,000 participants and three conditions). For each we quantified performance as the proportion of times the simulated participant chose the ‘correct’ high reward option in the relevant condition. Results showed that average performance improved as learning rates increased, up to learning rates of approximately 0.55 (Figure 2e). Crucially, this optimal alpha was above the highest learning rate shown by any of our participants in any recipient condition, meaning a higher learning rate on our task was associated with better performance.”

Figure 2e & f (page 11) -

“(e) Average percentages of correct choices (high probability of reward option) associated with 30,000 simulated α values (10,000 synthetic participants, 3 recipient conditions) show that an optimal learning rate is approximately 0.55 in this task. The range of α values for our participants was below this peak (grey shading), such that a higher learning rate was associated with better performance. (f) Correlation between percentage correct and learning rate across participants. There was a significant correlation between learning rate and accuracy ($r_{s(150)}=0.58$ [0.46, 0.68], $p<0.001$) (see Supplementary Table 3 for each separate age group and recipient combination; r_s in all cases > 0.46 , $ps <0.001$). Shaded area represents 95% confidence interval.”

Supplement (page 5) -

Supplementary Table 3. Correlations between learning rates and performance

	Young			Older		
	α_{self}	α_{other}	$\alpha_{no one}$	α_{self}	α_{other}	$\alpha_{no one}$
r_s	0.46	0.66	0.65	0.68	0.66	0.61
p	$<0.001^{***}$	$<0.001^{***}$	$<0.001^{***}$	$<0.001^{***}$	$<0.001^{***}$	$<0.001^{***}$

Note. Performance quantified as the percentage of trials on which participants choose the high reward option. Asterisks represent significance ($^{***}p<0.001$).

We have also updated the text to discuss the relationship between performance and learning rates (page 10):

“Higher learning rates are associated with better performance

As a final check of the robustness of our model and to enable clear interpretation of any differences in learning rate between recipients and age groups (c.f. Zhang et al., 2020), we conducted an additional simulation experiment. We simulated data from 10,000 participants using the $3\alpha 1\beta$ model. This created 30,000 values of α from the three recipient conditions, spanning the full range of possible values from 0 to 1 (see Methods). For each, we quantified the associated performance as the percentage of times the synthetic participant chose the high reward option, averaged across the blocks for the relevant recipient. Plotting the learning rates against performance (Figure 2e) shows that the optimal value of α is approximately 0.55. This is higher than all the values of α found on our task in any recipient condition for either age group. Therefore, higher learning rates were associated with better performance. We further established this link by correlating learning rates and performance in the empirical data from our participants. We found a strong correlation between learning rates and performance overall ($r_{s(150)}=0.58$ [0.46, 0.68], $p<0.001$; Figure 2f) and in each recipient and age group combination (Supplementary Table 3).”

R3.1b: In particular, the fact that learning rates were positively related to psychopathy in younger adults suggests that other processes besides prosocial concern were likely at play.

Response: We are sorry for any confusion, the association between learning rates and psychopathic traits in younger adults is not significant ($r_{s(74)}=0.21$ [-0.02, 0.42], $p=0.07$), even before correcting for multiple comparisons (FDR-corrected $p=0.22$) so should not be interpreted as positive. We have changed the wording describing this non-significant correlation and apologise that our original description of it having “a positive sign” could have been misinterpreted. This now reads (page 16):

“Intriguingly, this relationship was significantly more negative ($Z=3.28$, $p=0.001$) than the equivalent correlation in young adults, which was not significant ($r_{s(74)}=0.21$ [-0.02, 0.42], $p=0.07$; Figure 4b).”

We have also added the FDR-corrected p values in the legend for figure 4 (page 18):

“(a) For older adults, levels of psychopathic traits are negatively correlated with prosocial learning rates ($r_s=-0.33$ [-0.52, -0.11], $p=0.005$, false discovery rate (FDR) corrected $p=0.03$). (b) There is no significant relationship for young adults ($r_s=0.21$ [-0.02, 0.42], $p=0.07$, FDR-corrected $p=0.22$) and the correlation is significantly more negative ($Z=3.28$, $p=0.001$) for older than young adults.”

R3.1c: Relatedly, the discussion section suggests: “Our findings suggest that despite declines in learning ability associated with ageing, motivation could play a role in preserving learning to help others.”

However, this is not directly tested. The authors note that prosocial and self-oriented learning involve distinct neural mechanisms; as a result, another possibility is that mechanisms related to self-oriented learning decline whereas mechanisms related to other-oriented learning do not. This interpretation would not relate to prosocial motivation. Of

course, this finding might still have interesting implications given the importance of social connections in older adults, but would require a distinct interpretation.

Response: Thank you for the opportunity to discuss our interpretation further. We agree that one implication of distinct neural systems being involved in self and other learning might lead to self-learning declining and prosocial learning remaining intact. Since our use of the term 'motivation' in this context could be misinterpreted we have edited as follows (page 21):

“Our findings suggest that despite declines in learning ability associated with ageing, prosocial learning – learning to help others – is preserved. This finding aligns with an emerging literature showing older adults may be more prosocial and less self-biased than younger adults (Engel, 2011; Matsumoto et al., 2016; Mayr & Freund, 2020).”

R3.2a: 2. A second concern is that two central claims are written in ways that can easily be misunderstood. The abstract states: “Strikingly however, older adults showed reduced self-bias, with a relative increase in the rate at which they learnt about actions that helped others, compared to themselves. Moreover, we find evidence that these group differences are associated with changes in psychopathic traits over the lifespan.”

Although “a relative increase in the rate at which they learnt about actions that helped others” is technically true, this makes it sound like prosocial learning increased, whereas the primary finding is that self-oriented learning rates decreased.

Response: Thank you for this comment and the opportunity to clarify. We believe our primary finding is the relative change between self-relevant learning and prosocial learning ability between the two age groups. If learning in general was worse in older adults, we might predict it to affect both self and other-relevant learning. In this case, both self-relevant learning rates and prosocial learning rates should reduce. However, what we observe is that prosocial learning rates remain the same in the two groups, despite self-relevant learning rates changing. This is formally captured in a group * recipient interaction effect. However, we have removed the description described by the reviewer from the abstract and made sure we appropriately qualify the effect in all places in the manuscript.

The abstract now reads:

“Reinforcement learning is a fundamental mechanism displayed by many species. However, adaptive behaviour depends not only on learning about actions and outcomes that affect ourselves, but also those that affect others. Here, using computational reinforcement learning models, we tested whether young (age 18-36) and older (age 60-80, total n=152) adults can learn to gain rewards for themselves, another person (prosocial), or neither individual (control). Detailed model comparison showed that a model with separate learning rates for each recipient best explained behaviour. Young adults were faster to learn when their actions benefitted themselves, compared to helping others. Strikingly, compared to younger adults, older adults showed preserved prosocial learning rates but reduced self-relevant learning rates. Moreover, psychopathic traits were lower in older adults and negatively correlated with prosocial learning. These findings suggest learning how to benefit others is preserved across the lifespan with implications for reinforcement learning and theories of healthy ageing.”

We have also made edits to the discussion to qualify the effect:

Page 20 -

“However, this self-bias was reduced in older adults who showed a relative increase higher prosocial learning rates, relative to their own self-relevant learning rates, than young adults.”

Page 21 -

*“These findings suggest that the observed decline in self-relevant learning rates, but relative-increase **preservation** of prosocial learning rates”*

Page 23 -

*“Thus, the ~~relative~~ increase in prosocial learning rates, **relative to self-relevant learning rates**, suggests older adults are reinforced by outcomes for others and themselves more similarly than younger adults”*

R3.2b: Similarly, “these group differences are associated with changes in psychopathic traits over the lifespan” sounds very much like it is describing a mediation model where psychopathic traits change, and this change is associated with group differences in learning rates. However, this is not the pattern of results described in the paper; group differences in learning rates were not associated with group differences in psychopathic traits. Instead, psychopathic traits had different patterns of correlation with learning rates within groups.

Response: Thank you for highlighting this and we apologise if our previous wording implied mediation effects. Your comment highlighted the fact that our observation of a correlation between prosocial learning rates and psychopathic traits for older but not younger adults may suggest a mediation effect, thank you for this. We have now run this additional analysis, using the difference between $\alpha_{\text{other}} - \alpha_{\text{self}}$ as the outcome variable as this difference significantly interacted with age group in our main model. We term this difference ‘relative prosocial learning rate’ to aid readability. As outlined below, a standard mediation model testing whether psychopathic traits mediated the effect of age group on relative prosocial learning rate did not show evidence of a mediation and we have changed the sentence in the abstract that you highlighted. However, as would be predicted if psychopathic traits explain prosocial learning in older, but not younger, adults we found evidence of a moderated mediation. We have included details on the method for and results of both these mediation models in the revised paper (see below).

Methods (page 34) -

“For the mediation analysis, we used the mediate function (mediation package (Tingley et al., 2014)) combined with robust linear models (rlm function, MASS package (Venables & Ripley, 2002)). This method estimates the unstandardised indirect effects, with 95% confidence intervals and significance, through a bootstrapping procedure with 10,000 bootstrapped samples. The outcome in the mediation model was relative prosocial learning rate ($\alpha_{\text{other}} - \alpha_{\text{self}}$), the predictor was age group, and the mediator was psychopathic traits on the core affective-interpersonal subscale of the SRP-IV-SF. We calculated two mediation models. The first, a standard mediation model, included an indirect path from age group to relative prosocial learning rate via psychopathic traits as a mediator (Figure 5a). The second included an interaction between age group and psychopathic traits in predicting relative prosocial learning rate to allow the possibility of a moderated mediation (Imai

et al., 2010). Specifically, this type of moderated mediation examines whether the effect of the mediator (in this case psychopathic traits) on the outcome ($\alpha_{other} - \alpha_{self}$) is moderated by the predictor (age group) (Judd & Kenny, 1981; Preacher et al., 2007). In other words, psychopathic traits could be a mediator for one age group but not the other (Figure 5b).

Results (page 16-17) -

“This pattern of results was the same when correlating ‘relative prosocial learning rate’ (the difference between $\alpha_{other} - \alpha_{self}$) with core psychopathic traits.

[...]

Given the age-group differences in levels of psychopathic traits and their correlations with prosocial learning, our final analysis considered whether scores on the core affective-interpersonal psychopathic traits measure mediated the effect of age group on relative prosocial learning rate ($\alpha_{other} - \alpha_{self}$). A standard mediation model (Figure 5a) did not show evidence for a significant mediation. However, as would be predicted if the link between psychopathic traits and prosocial learning depends on age, we found evidence for a moderated mediation. This revealed core psychopathic traits mediated the effect of age group on relative prosocial learning rate for older adults (unstandardised indirect effect=0.006 [0.001, 0.013], $p=0.008$, proportion mediated=0.32) but not for young adults (unstandardised indirect effect=-0.001 [-0.007, 0.006], $p=0.65$; see Figure 5b for standardised coefficients). In summary, young and older adults differed in levels of psychopathic traits and whether or not psychopathic trait scores explained the extent to which their learning rates were relatively prosocial or self-biased.”

Figure 5 (page 19) -

“Figure 5. Psychopathic traits mediate the effect of age group on relative prosocial learning ($\alpha_{other} - \alpha_{self}$) for older adults only. (a) A standard mediation model does not show evidence that psychopathic traits mediate the effect of age group on relative prosocial learning. Although accounting for psychopathic traits means the significant direct effect of age group on relative prosocial learning (standardised coefficient=0.28, $p=0.04$) becomes non-significant (standardised coefficient=0.23, $p=0.11$), psychopathic traits do not predict relative prosocial learning overall so there is no mediation. (b) Evidence of a moderated mediation is revealed when accounting for differences between young and older adults in how psychopathic traits predict relative prosocial learning. Psychopathic traits are a

mediator for older but not younger adults. Psychopathic traits are scores on the affective-interpersonal subscale of the Self Report Psychopathy scale, Y: young, O: older, asterisks represent significant effects ($p < 0.05$, ** $p < 0.01$).*"

We have not included the mediation in the abstract due to space constraints, the abstract now reads:

"Reinforcement learning is a fundamental mechanism displayed by many species. However, adaptive behaviour depends not only on learning about actions and outcomes that affect ourselves, but also those that affect others. Here, using computational reinforcement learning models, we tested whether young (age 18-36) and older (age 60-80, total $n=152$) adults can learn to gain rewards for themselves, another person (prosocial), or neither individual (control). Detailed model comparison showed that a model with separate learning rates for each recipient best explained behaviour. Young adults were faster to learn when their actions benefitted themselves, compared to helping others. Strikingly, compared to younger adults, older adults showed preserved prosocial learning rates but reduced self-relevant learning rates. Moreover, psychopathic traits were lower in older adults and negatively correlated with prosocial learning. These findings suggest learning how to benefit others is preserved across the lifespan with implications for reinforcement learning and theories of healthy ageing."

We have also edited the related sentence summarising the correlations with psychopathic traits in the introduction (page 6):

"Older adults had significantly reduced levels of psychopathic traits compared to younger adults and in older adults, lower psychopathic trait scores correlated with prosocial learning rates (Figure 4a & b). These effects were not explained by individual differences in IQ, memory or attention abilities."

R3.3: 3) The primary analyses examine the interaction of age x target across self and other. However, it's not obvious to me why the analysis should be restricted to these two targets, instead of examining the trend across all three conditions. That is, is there less of a reduction in learning rate for "other" than for "no one" among older adults? This interaction would be important to inform whether results are truly due to changes in prosocial learning as opposed to changes in self-learning mechanisms alone.

Response: We apologise for any lack of clarity that has led to the incorrect interpretation that the no one condition was not included in our analysis. Our main results come from a model that includes age group (young, older) and recipient and the age group * recipient interaction, with all levels of the recipient condition (self, other, no one).

Our motivation for using a robust linear mixed-effects model (RLMM) was that this approach does not rely on parametric assumptions and accounts for the within-subject nature of the recipient manipulation. Additionally, unlike an anova model, this model generates coefficients (with confidence intervals and significance values) for terms that compare pairs of factor levels (for example self vs. other) (Bates et al., 2015). As the format of the results generated by our RLMM is slightly different to an anova model we additionally ran the same model through a method that treats it as a robust two-way anova (mixed between-within subject design) using a bootstrapping method to test the overall group*recipient interaction (sspbi function from the WRS2 package (Mair & Wilcox, 2020), 10,000 iterations). This shows a significant overall interaction ($p = 0.045$). We have not currently added this to the

paper as it represents the same analysis as our RLMM and cannot be interpreted without the age * pairs of recipient conditions interactions and the post hoc comparisons we report. However, we would be happy to include it if you think it would be useful for readers. We have highlighted the rationale of our robust linear mixed-effects approach in the methods section.

In all analyses using any form of linear model and a predictor with more than 2 levels, the number of these terms (for both main effects and interactions) is always the number of levels – 1 (Fox, 2015), so 2 in our case. These are self vs. other and self vs. no one because the self condition is the reference group. Crucially, we also now report the model interaction term for the non-significant age group * recipient [self vs. no one] interaction, which we recognise should have been included before. Combined with the significant age group * recipient [self vs. other] interaction, this shows prosocial learning, relative to learning for the self, differs between the age groups in a way that learning for no one, relative to learning for the self, does not. We consider the difference between the other and no one conditions through post hoc comparisons for each age group separately and do not use wording that suggests a formal interaction when interpreting the results. We have edited the wording in the results section to better describe the fact that we consider the pattern within-subject and between-subject comparisons together in generating our conclusions.

Following on from the significant interaction of recipient [self vs other] and age, we describe prosocial learning as ‘preserved’ in older adults because our between-group comparison showed strong Bayesian evidence in support of no difference between the age groups in prosocial learning rates. In contrast, learning rates for the self were significantly lower in older, compared to younger adults. A decrease in learning rates during self-relevant learning with age has been shown in multiple previous studies, as outlined in our introduction. Against this background of robust evidence that self-relevant learning is disrupted with age, we believe the preservation of prosocial learning rates in older adults at the same level as young adults is striking. The comparison of learning rates for no one between the age groups did not show a significant difference or evidence for no difference, so we do not interpret this as the same or different between young and older adults. We have rewritten the main part of our results section, which now reads (page 13-14):

*“We analysed the condition-specific learning rates from our best fitting computational model using a robust linear mixed-effects model (RLMM; see Methods). The RLMM fixed effects were age group (young, older), recipient (self, other, no one), as well as the age group * recipient interaction. While this RLMM includes all three recipient conditions, it generates coefficients (main effect and the interaction with age group) contrasting pairs of recipient conditions – [self vs. other] and [self vs. no one]. These are more interpretable than an omnibus test that would not show which recipient conditions were driving an effect or interaction. We followed up these results with planned comparisons, between the older and younger group in each recipient condition, and between pairs of recipient conditions within each age group.*

*Across age groups, participants showed a higher learning rate when rewards were for themselves, compared to for another person (recipient [self vs. other]: $b=-0.024$ [-0.034, -0.014], $z=-4.79$, $p<0.001$). Importantly however, this pattern differed between age groups. The difference between learning rates for self and other was reduced in older compared to younger adults (recipient [self vs. other] * age group interaction: $b=0.016$ [0.002, 0.030], $z=2.29$, $p=0.02$). Between-group comparisons showed older*

adults learnt more slowly for themselves compared to younger adults ($W=3512$, $Z=-2.63$, $r_{(150)}=0.22$ [0.06, 0.36], $p=0.009$). However, prosocial learning was preserved, with a Bayes factor suggesting strong evidence of no difference in α_{other} between young and older adults ($W=3042$, $Z=-0.86$, $r_{(150)}=0.07$ [0.00, 0.24], $p=0.39$, $BF_{01}=4.26$). Within-subject comparisons of α_{self} and α_{other} in each age group showed that young adults had higher learning rates for themselves, relative to another person ($V=659$, $Z=-4.04$, $r_{(75)}=0.47$ [0.26, 0.63], $p<0.001$). In contrast, older adults showed no significant difference between learning rates for self and other ($V=1150$, $Z=-1.45$, $r_{(75)}=0.17$ [0.01, 0.38], $p=0.15$, $BF_{01}=1.08$).

As expected, across age groups learning was slower for no one than self (recipient [self vs. no one]: $b=-0.023$ [-0.033, -0.013], $z=-4.57$, $p<0.001$). Unlike α_{self} vs. α_{other} , learning for self compared to no one did not interact with age group (recipient [self vs. no one] * age group interaction: $b=0.008$ [-0.006, 0.022], $z=1.15$, $p=0.25$). Within-subject comparisons between α_{self} and $\alpha_{no\ one}$ in each age group showed that both groups learnt preferentially for themselves compared to no one (young adults: $V=928$, $Z=-2.62$, $r_{(75)}=0.30$ [0.07, 0.51], $p=0.009$; older adults $V=901$, $Z=-2.76$, $r_{(75)}=0.32$ [0.09, 0.53], $p=0.006$). There was no significant difference between the age groups in $\alpha_{no\ one}$ but also no evidence in support of the null ($W=3241$, $Z=-1.61$, $r_{(150)}=0.13$ [0.01, 0.29], $p=0.11$, $BF_{01}=2.04$).

Considering differences in learning between α_{other} and $\alpha_{no\ one}$, young adults did not differentiate between another person and no one, with strong Bayesian evidence for no difference ($V=1533$, $Z=-0.57$, $r_{(75)}=0.07$ [0.00, 0.31], $p=0.57$, $BF_{01}=5.08$). In contrast, older adults had higher learning rates for another person, compared to no one ($V=976$, $Z=-2.37$, $r_{(75)}=0.27$ [0.05, 0.49], $p=0.02$). Crucially, this shows that older adults' lack of differentiation between self and other was not simply because they were insensitive to the recipient condition.

Finally, we also observed an effect of age on both learning rates overall and temperature parameters. Older adults showed slower learning overall compared to younger adults ($b=-0.019$ [-0.028, -0.009], $Z=-3.73$, $p<0.001$) and higher levels of exploration of choice options (median β young: 0.05, older: 0.19, $W=1511$, $Z=-4.89$, $r_{(150)}=0.40$ [0.26, 0.53], $p<0.001$; Supplementary Figure 1).

In summary, older adults prosocial learning was preserved at the same rate as young adults, despite age-related declines in self-relevant learning rates. In other words, young adults showed a self-bias in learning, but older adults distinguished between themselves and others significantly less than the young participants. Only older adults, not young adults, distinguished between rewards for another person and no one.”

Methods (page 32-33) -

“We used a robust linear mixed-effect model (RLMM; `rlmer` function; `robustlmm` package (Koller, 2016) to predict learning rates and generalised linear mixed-effects

models (glmer function; lme4 package (Bates et al., 2015)) for the trial-by-trial data (binary outcome of choosing the high vs. low reward option). We used (robust / generalised) linear mixed-effects models as these account for the within-subject nature of the recipient manipulation and do not rely on parametric assumptions. Additionally, unlike an anova model or omnibus test, an RLMM generates coefficients (with confidence intervals and significance values) for terms that compare pairs of factor levels (for example self vs. other). This approach is more informative than an anova when the factor has more than two levels, which is our case as the recipient factor has three levels (self, other, no one)."

Minor points:

R3.4a: 1. The introduction states: "It is largely unknown how ageing affects social functioning, despite the critical importance of this question." This is too broad a claim and should more specifically address prosocial learning in aging.

Response: We apologise for our lack of clarity here and we have amended our wording as follows (page 3):

"Senescence is associated with a multitude of changes including declines in cognitive functioning and perception, but perhaps preservation of affective processing and social cognitive abilities (Blanchard-Fields, 2007; Blanchard-Fields & Hess, 1999; Samanez-Larkin & Knutson, 2015). However, less is known about how ageing affects social behaviour, despite the critical importance of this question. Social isolation has been found to be as damaging to physical health as smoking or excessive drinking (Holt-Lunstad et al., 2015). Social behaviours that benefit others – prosocial behaviours – are vital for maintaining social bonds and relationships (Fehr & Fischbacher, 2003) across the lifespan. In addition to the benefits for others, prosociality has been linked to improved life satisfaction (Buchanan & Bardi, 2010), mental wellbeing (Raposa et al., 2016), and physical health (Post, 2005) for the person being prosocial, all of which could contribute to healthy ageing. A key aspect of prosocial behaviour is the ability to learn associations between our own actions and outcomes for other people (Lockwood et al., 2016)."

R3.4b: The authors might also elaborate on how prosocial learning contributes to our understanding of social functioning in aging.

Response: Thank you for the opportunity to elaborate further. We have added the following in the introduction (page 4):

"Alternatively, prosocial learning may depend not only on our learning ability, but also our motivation to help others. Results from experiments using economic games to measure prosociality have found that older adults tend to be more generous (Engel, 2011; Matsumoto et al., 2016). There is also evidence of an age-related increase in charitable donations to individuals in need (Sze et al., 2012). At work, older adults engage in more prosocial behaviours than younger adults, according to both self-report data and colleagues' ratings (Ng & Feldman, 2008). Finally, self-reported altruism and decisions to donate to others have been shown to increase with age (Hubbard et al., 2016). However, one limitation of these studies is that the paradigms often place self and other reward preferences in conflict. Money for the other person depends on less money for oneself. Moreover, older adults generally have higher accumulated wealth, which would be an important confound in studies of monetary

exchange (Cheung & Lucas, 2015). Prosocial learning avoids this confound by separating outcomes for oneself from outcomes for others. If older adults do indeed value outcomes for others more than young adults, we might expect that whilst self-relevant learning declines with ageing, prosocial learning could be preserved. Comparing young and older adults on self-relevant and prosocial learning provides an opportunity to dissociate how possible age-related changes in cognitive ability and social behaviour impact on learning.”

R3.5: 2. When describing overall group differences in psychopathy, it would help if the authors can report the means directly in the text.

Response: Thank you for this suggestion, we have now added the means in the text (page 16):

“Comparing the two age groups on these scales showed that older participants had significantly lower scores than young participants on both the core affective-interpersonal (young mean=24.36, older mean=21.09, $W=3558$, $Z=-3.15$, $r_{(148)}=0.26$ [0.11, 0.41], $p=0.002$) and the lifestyle-antisocial subscales (young mean=22.89, older mean=20.27, $W=3471$, $Z=-2.82$, $r_{(148)}=0.23$ [0.07, 0.38], $p=0.005$).”

R3.6: 3. Fig 3a appears to show far less variability in other-learning rates relative to self or no one. How do the authors interpret this?

Response: Thank you for this observation. We ran additional analyses to quantitatively test for differences in variability between the learning conditions. Levene’s tests comparing variance in the learning rates for each pair of recipient conditions showed significant differences between self and other, other and no one, but also between self and no one (all $ps < 0.001$). Therefore, all learning conditions have different levels of variability from one another and not only when learning for other relative to self and no one. We note that we used nonparametric tests that account for any heterogeneity of variance between learning conditions throughout.

Reviewers' comments:

Reviewer #1 (Remarks to the Author):

The authors have provided reasonable responses to all of my earlier comments. I still have concerns about saying the effects are partially moderated by psychopathic traits in the abstract and throughout - but maybe that's just personal preference. I think it would be better to be more specific about what is accounting for the effects, which seems to be more specific to sensitivity to social cues and concern for others. Psychopathic traits still makes it sound like the younger people are more likely to be psychopaths to me, rather than emphasizing the specific potential mechanisms of the effects.

Gregory Samanez-Larkin

Reviewer #2 (Remarks to the Author):

The authors have provided a thoughtful, detailed, and clear revision that has strengthened the manuscript significantly. I appreciate the additional analyses and methodological clarification, as well as the more nuanced interpretation and embedding of the results.

All my previous comments were addressed and I have no additional comments. Thank you again for this clear revision.

Reviewer #3 (Remarks to the Author):

I have carefully read the authors' response letter and revisions, and I found them to be incredibly thorough and thoughtful--well done. The clarifications and additional analyses addressed all concerns I had, and I believe the paper can be published in its current form.

Reviewers' comments:

Reviewer #1 (Remarks to the Author):

The authors have provided reasonable responses to all of my earlier comments. I still have concerns about saying the effects are partially moderated by psychopathic traits in the abstract and throughout - but maybe that's just personal preference. I think it would be better to be more specific about what is accounting for the effects, which seems to be more specific to sensitivity to social cues and concern for others. Psychopathic traits still makes it sound like the younger people are more likely to be psychopaths to me, rather than emphasizing the specific potential mechanisms of the effects.

Gregory Samanez-Larkin

Response: Thank you so much for your time reviewing our work and comments that have improved the paper. We have now further revised the wording around psychopathic traits throughout the manuscript, in particular adding 'subclinical' and 'self-reported' throughout to make it clear we are not implying the young participants are more likely to be psychopaths in diagnostic or clinical terms. We have also included '(lack of concern for others)' in the abstract and introduction as you suggested.

We believe it is important to keep the term 'psychopathic traits', with these additional terms alongside it, throughout the paper as 'psychopathic traits' most accurately describes what we measure with the Self Report Psychopathy scale (Paulhus et al., 2017), a robust and validated self-report measure of subclinical psychopathic traits based on the gold standard Hare Psychopathy Checklist Revised (see <https://storefront.mhs.com/collections/srp-4>). Our analysis focuses on the affective-interpersonal subscale of this measure, so we also use this specific term and now better define this subscale as measuring traits relevant to concern for others. Research into psychopathy and psychopathic traits is an established field of research, with a corresponding society – the Society for the Scientific Study of Psychopathy – which defines psychopathy as “a constellation of traits” (<https://psychopathysociety.org/page/AboutPsychopathy>). Moreover, the recognition that these traits exist in low subclinical levels in the general population is no different from subclinical traits associated with other mental disorders such anxiety and depression existing in the general population.

In addition to these changes to how we introduce and define the self-report psychopathic traits measure, we have also revised our wording around the mediation analysis to make it clear that we are not claiming our effects are partially moderated by psychopathic traits. We note that we do not mention this analysis in the abstract and only highlight the correlations. The results section on this analysis now focuses on 'indirect effects' of age group on relative prosocial learning through self-reported psychopathic traits. We appreciate your recognition that people have different preferences on these analyses and hope that our revised wording better accounts for these different preferences.

Reviewer #2 (Remarks to the Author):

The authors have provided a thoughtful, detailed, and clear revision that has strengthened the manuscript significantly. I appreciate the additional analyses and methodological clarification, as well as the more nuanced interpretation and embedding of the results.

All my previous comments were addressed and I have no additional comments. Thank you again for this clear revision.

Response: We really appreciate your time reviewing our manuscript and positive feedback on our revisions, as well as your comments that helped us improve the paper, thank you.

Reviewer #3 (Remarks to the Author):

I have carefully read the authors' response letter and revisions, and I found them to be incredibly thorough and thoughtful--well done. The clarifications and additional analyses addressed all concerns I had, and I believe the paper can be published in its current form.

Response: We really appreciate your time reviewing our manuscript and positive feedback on our revisions, as well as your comments that helped us improve the paper, thank you.